# New Insights of Geomorphologic and Lithologic Features on Wudalianchi Volcanoes in the Northeastern China from the ASTER Multispectral Data

**Han Fu [1,2], Bihong Fu [1,*], Yoshiki Ninomiya [3] and Pilong Shi [1]**

[1] Aerospace Information Research Institute, Chinese Academy of Sciences, Beijing 100094, China; fuhan2017@radi.ac.cn (H.F.); shipl@radi.ac.cn (P.S.)

[2] University of Chinese Academy of Sciences, Beijing 100094, China

[3] Geological Survey of Japan, National Institute of Advanced Industrial Science and Technology, Higashi 1-1-1, Tsukuba 305-8567, Japan; yoshiki.ninomiya@aist.go.jp

[*] Correspondence: fubh@radi.ac.cn; Tel.: +86-10-8217-8096

**Abstract:** Advanced Spaceborne Thermal Emission and Reflection Radiometer (ASTER) imaging system onboard NASA's (National Aeronautics and Space Administration's) Terra satellite is capable of measuring multispectral reflectance of the earth's surface targets in visible and infrared (VNIR) to shortwave infrared (SWIR) (until 2006) as well as multispectral thermal infrared (TIR) regions. ASTER VNIR stereo imaging technique can provide high-resolution digital elevation models (DEMs) data. The DEMs data, three-dimensional (3D) perspective, and ratio images produced from the ASTER multispectral data are employed to analyze the geomorphologic and lithologic features of Wudalianchi volcanoes in the northeastern China. Our results indicate that the 14 major conical volcanic craters of Wudalianchi volcanoes are arranged as three sub-parallel zones, extending in a NE (Northeast) direction, which is similar to the direction of regional fault system based on the ASTER DEMs data. Among the 14 volcanic craters in Wudalianchi, the Laoheishan, and Huoshaoshan lavas flows, after the historic eruptions, pouring down from the crater, partially blocked the Baihe River, which forms the Five Large Connected Pools, known as the Wudalianchi Lake. Lithologic mapping shows that ASTER multispectral ratio imagery, particularly, the Lava Flow Index (LFI) (LFI = B10/B12) imagery, can clearly distinguish different lava flow units, and at least four stages of volcanic eruptions are revealed in the Wudalianchi Quaternary volcano cluster. Thus, ASTER multispectral TIR data can be used to determine relative dating of Quaternary volcanoes in the semi-arid region. Moreover, ASTER 3D perspective image can present an excellent view for tracking the flow directions of different lavas of Wudalianchi Holocene volcanoes.

**Keywords:** 3D perspective view; morphology; lithology; Wudalianchi volcano; ASTER multispectral data

## 1. Introduction

Remote sensing can provide a certain resolution of spectral, spatial, and temporal coverage based on the type of sensor for geologic mapping and monitoring at numerous volcanoes throughout the world [1,2]. The use of spaceborne and airborne remote sensing data to monitor the volcanoes and map the products of eruptions has been ongoing for decades [3]. Satellite remote sensing data have been widely used to detect or monitor the eruption of volcanoes [4–6]. However, how to document the products of volcanic eruptions and morphology using satellite remote sensing data is another challenge for geoscientists [6,7]. The weathering of active lava flows in arid and semi-arid environments is

accompanied by changes in their thermal infrared emittance spectra [8,9]. The spectral differences caused by the weathering can be measured and mapped with multispectral imaging system [8].

The Advanced Spaceborne Thermal Emission and Reflection Radiometer (ASTER) sensor was launched in 18 December, 1999, onboard the first NASA's Earth Observation System (EOS) series of satellites, Terra [10,11]. The ASTER covered a wide spectral region with 14 bands from visible to thermal infrared with high spatial, spectral and radiometric resolution as shown in Table 1. Three visible and near infrared (VNIR) bands, six shortwave infrared (SWIR) bands (until 2006), and five thermal infrared (TIR) bands with the spatial resolution of 15 m, 30 m, and 90 m, respectively. In addition, the bands 3N and 3B in near infrared bands have a stereoscopic capability, which can be used to generate DEM [12]. ASTER-TIR is the first satellite-borne multispectral TIR remote sensing system with spectral, spatial, and radiometric resolutions adequate to be used for geologic applications, such as determining the relative age dating of lavas [13,14]. Compared with two bands of Landsat TM or ETM in the SWIR region (between 1.6 to 2.5 microns), the ASTER SWIR sensor has six bands in this region and has the capability to identify mineral component of surface rocks in the semi-arid to arid region [15,16]. The ASTER multispectral SWIR and TIR sensors can provide an important tool for monitoring heat flow related to volcanic activities [4,5]. Therefore, ASTER can provide a potential tool for mapping the products from active volcanoes from regional to global scales.

**Table 1.** Wavelength Range of The Advanced Spaceborne Thermal Emission and Reflection Radiometer Data [10,12].

| Band | Wavelength Range (μm) | Band Type | Spatial Resolution (m) |
|------|------------------------|-----------|-------------------------|
| B1   | 0.52–0.60              | VNIR (visible and near infrared) | 15 |
| B2   | 0.63–0.69              | VNIR      | 15 |
| B3N  | 0.76–0.86              | VNIR      | 15 |
| B3B  | 0.76–0.86              | VNIR      | 15 |
| B4   | 1.60–1.70              | SWIR (shortwave infrared) | 30 |
| B5   | 2.145–2.185            | SWIR      | 30 |
| B6   | 2.185–2.225            | SWIR      | 30 |
| B7   | 2.235–2.285            | SWIR      | 30 |
| B8   | 2.295–2.365            | SWIR      | 30 |
| B9   | 2.36–2.43              | SWIR      | 30 |
| B10  | 8.125–8.475            | TIR (thermal infrared) | 90 |
| B11  | 8.475–8.825            | TIR       | 90 |
| B12  | 8.925–9.275            | TIR       | 90 |
| B13  | 10.25–10.95            | TIR       | 90 |
| B14  | 10.95–11.65            | TIR       | 90 |

There are two Quaternary volcanoes with historic eruptions in northeastern China (Figure 1a, WD, Wudalianchi Volcano; TC, Changbaishan Tianchi Volcano). Petrology, chronology, and geochemistry of these Quaternary volcanoes have been widely addressed by researchers [17–20]. However, general research on the topographic and geomorphologic features of these Quaternary volcanoes is still lacks. Furthermore, no detailed map shows the distribution of lava flows from these different stages.

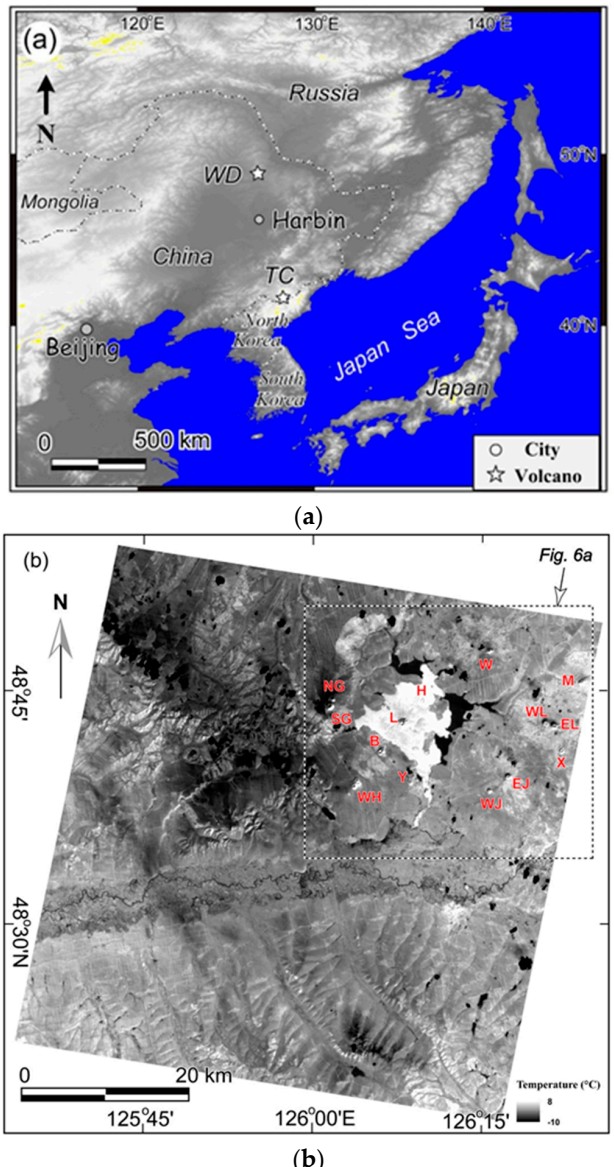

**Figure 1.** (**a**) Topographic map generated from the USGS (United States Geological Survey) SRTM (Shuttle Radar Topography Mission) DEM (Digital Elevation Model) data, showing the locations of Wudalianchi (WD) and Tianchi (TC) volcanoes in northeast China. (**b**) ASTER thermal infrared (band 13) image observed at 6 April, 2004 in the Wudalianchi volcanic region. WH: Wohushan; SG: South Gelaqiushan; NG: North Gelaqiushan; B: Bijiashan; L: Laoheishan; H: Huoshaoshan; W: Weishan; WJ: West Jiaodebushan; EJ: East Jiaodebushan; X: Xiaogushan; WL: West Longmenshan; EL: East Longmenshan; M: Molabushan; Y: Yaoquanshan.

In this study, we documented the products of eruptions and morphology of Quaternary volcanoes in northeast China using satellite remote sensing techniques. The main goal of this study is to verify how to map geomorphologic features and lava flows of Quaternary volcanoes using ASTER multispectral data. A field investigation was conducted around the Wudalianchi volcanoes from late July to early August, 2005 as part of a collaborative project between the Institute of Geology and Geophysics, Chinese Academy of Sciences and Geological Survey of Japan, The National Institute of Advanced Industrial Science and Technology. During the field investigation, we observed the difference of vegetation coverage in the Wudalianchi volcanoes, which can provide useful information to understand the spectral difference of the lava flows in the Laoheishan and Huoshaoshan lava flow regions.

## 2. Geologic Setting

The Wudalianchi and Changbaishan volcanoes in northeastern China have been widely studied [18–22]. Previous studies indicated that they are intraplate volcanoes far from the West Pacific Plate subduction zone [22–24].

The Wudalianchi volcanic cluster is located on Wudalianchi city, about 350 km north of Harbin, capital of Heilongjiang Province (Figure 1a). The volcanic group covers an area of about 800 km$^2$ and is composed of 14 major volcanic cones (Figure 1b). These lava rocks in the Wudalianchi region are strong alkaline potassium-rich volcanic rock with an average K (Potassium) content of 5.28% and average SiO$_2$ content of 50.46%. [17,25].

The K-Ar (Potassium-Argon) dating of Wudalianchi monogenetic volcanic products suggested three stages of eruptions: Early-middle Pleistocene (circa 1.33 ± 0.08 to 0.8 ± 0.02 Ma), Late Pleistocene (circa 0.63–0.3 Ma), and historic periods [18,26]. Among the 14 volcanoes in the Wudalianchi volcanic cluster, the lava flows distributed around Xiaogushan (X), Yaoquanshan (Y), Wohushan (WH), and West Jiaodebushan (WJ) are alkaline basalt that belongs to the Early-middle Pleistocene; the lava flows from Molabushan (M), Bijiashan (B), South Gelaqiushan (SG), North Gelaqiushan (NG), Weishan (W), and East Jiaodebushan (EJ) are alkaline basalt that belongs to the Late Pleistocene [27]. However, it is still a big debate on the historic volcanic activity. Some researchers indicated that the most recent major explosive eruption occurred between 1719–1721 AD and formed craters of the Laoheishan (L) and Huoshaoshan (H) volcanoes [17,28]. Historical local documents suggest that the lava flows from Laoheishan and Huoshaoshan craters were mainly occurred in 1719–1721 AD and 1776 AD [26,27,29]. Tectonically, the Wudalianchi volcanic cluster is located in the triangular area formed by the three structural units of the Greater Khingan uplift, the Lesser Khingan uplift and the Songliao faulted basin. Previous studies suggested that the distribution of 14 volcanoes in the Wudalianchi area is associated with the Northeast (NE)-striking faults, Northwest (NW)-striking faults and near Eastwest (EW)-striking faults or fracture zones [30,31]. However, the geological interpretation of ASTER DEM image (Figure 2) shows that the NE and NW-striking faults are major structure in the Wudalianchi volcanic field. The most of volcanoes are extending as NE-striking direction (such as L, B, M and WH in Figure 2). We inferred that these conjugate faults might provide the pathway for magma migration during the volcanic eruption.

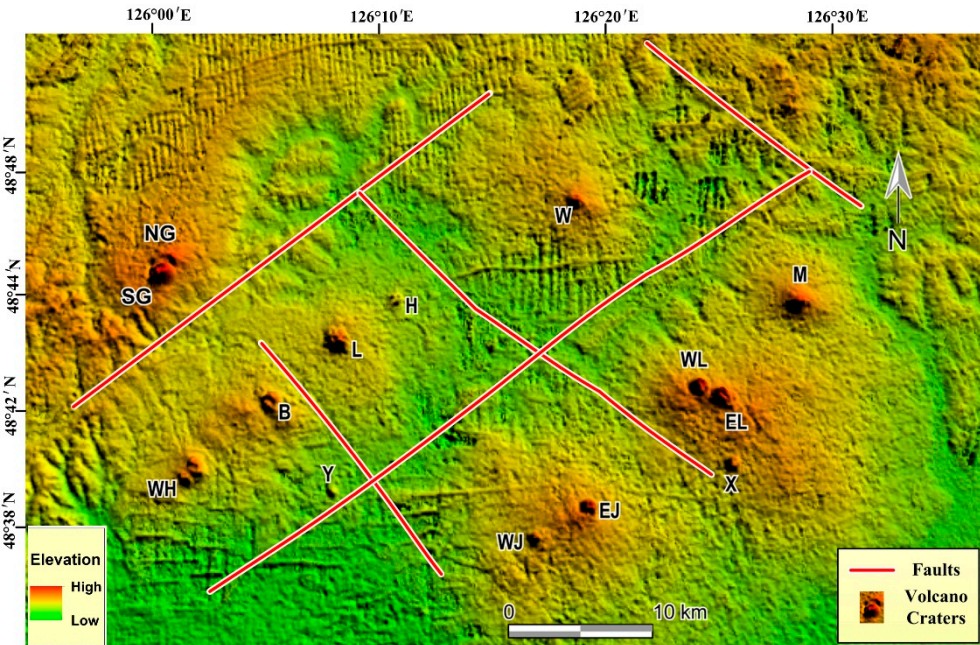

**Figure 2.** Geological sketch showing the relation between tectonic context and volcanism in the Wudalianchi region. Letters refer to name of volcanic craters are same as Figure 1b.

Concerning the geodynamics of the formation of the Wudalianchi volcanoes, some studies suggested that they are the intraplate volcano associated with the westward extending subduction slab of the West Pacific Plate [22–24]. Based on analysis of the topographic image as well as the previous data given by seismic evidence [32,33], we have drawn a three-dimensional geodynamic sketch map using Artificial Intelligence (AI) software, which shows the relationship between the active intraplate volcanoes in NE China and the deep subduction of the Pacific slab as shown in Figure 3. This geodynamic sketch map implies that the intraplate volcanoes in the interior of the Asian continent are not the back-arc volcanoes related to the subducting Pacific slab such as the Japanese Islands, but the continental volcanoes are likely induced by the deep subduction and dehydration of the west Pacific stagnant slab, possibly through hot and wet upwelling in the big mantle wedge under the NE China as suggested by the geophysical studies [34–37].

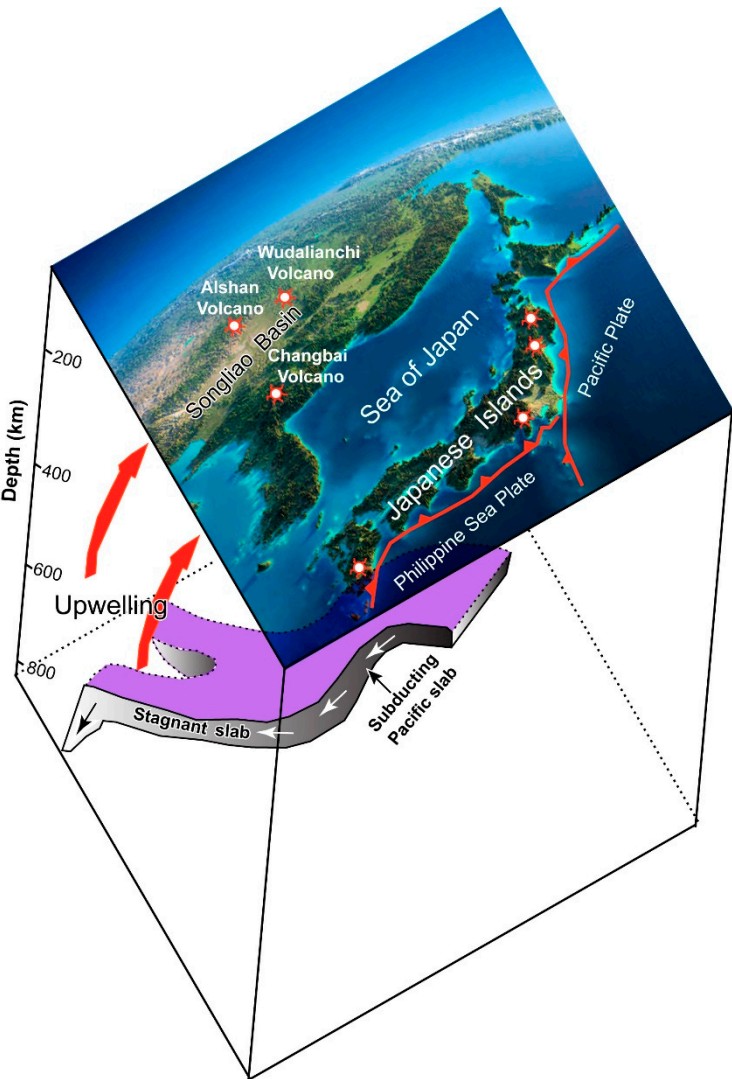

**Figure 3.** The three-dimensional (3D) geodynamic sketch map showing that the active volcanoes in Northeastern Asia are induced by the deep subduction of the western Pacific stagnant and subducting slabs.

The Wudalianchi volcano was successfully selected as one of the UNESCO (United Nations Educational, Scientific and Cultural Organization) Global Geoparks in 2003. The major reason is that the lava flow composition of the Wudalianchi volcanic belt is very special and is characterized by strong alkaline potassium-rich volcanic rock [38]. Its color ratio (40%–55%) is higher than the coarse

porphyry and ring rock (35%–20%), while at the same time, it does not contain basic plagioclase, which basalt should have; thus, it is neither a rough rock nor an alkaline basalt.

Moreover, Laoheishan and Huoshaoshan are the latest volcanoes in the Wudalianchi volcanic group, whose most recent eruptions occurred in 1719–1721 AD and 1776 AD as recorded by historic documents. Thus, their lava flows formed by the eruptions are bare, well preserved, and the lava flows characteristics are clear. As the geomorphologic and lithologic features of the eruptions on Laoheishan and Huoshaoshan are well preserved, the Wudalianchi volcano has the reputation of "Volcano Natural Museum" [39,40], which is another important reason for its selection in the UNESCO Global Geoparks.

## 3. Methodology

Three scenes of relatively cloud-free ASTER data covering the Wudalinchi region were used in this study. The DEM data, which belongs to the ASTER Level-3A product, were derived from the ASTER Level-1A data, with a vertical accuracy of 20 m [10,12]. See Table 2 for details. The datasets used in this study is described in a flowchart (Figure 4).

**Table 2.** Details of ASTER data.

| Obtained Date | Central Point | Product Level | Vertical Accuracy of Digital Elevation Model |
|---|---|---|---|
| 7 March, 2002 | 48.87°N, 126.42°E | orthorectified Level-3A | 20 m |
| 6 September, 2002 | 48.73°N, 125.82°E | orthorectified Level-3A | 20 m |
| 6 April, 2004 | 48.60°N, 125.98°E | orthorectified Level-3A | 20 m |

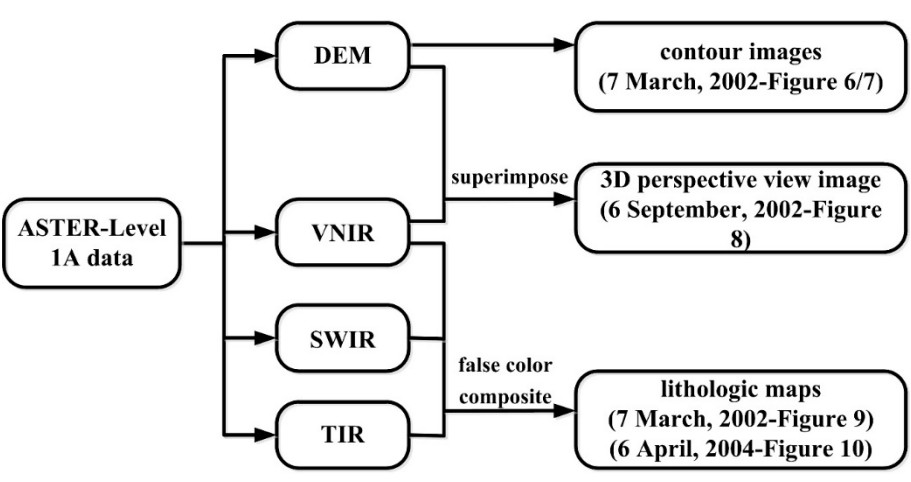

**Figure 4.** The flowchart of the datasets used in this study.

### 3.1. The Generation of Contour Image and 3D Perspective View Image

A contour image was derived automatically from ASTER Level-3A DEM data by using ER-Mapper (Earth Resource-Mapper) software. Three-dimensional (3D) perspective view image was generated through superimposing a three bands color composite image (the bands 2, 3, and 1 assigned in red, green, and blue, respectively) on the ASTER DEM data by using the same software.

### 3.2. Laboratory Measurement for the Emissivity of Lava Fows

The laboratory emissivity of typical lava rocks from the Wudalianchi region were measured by using portable FTIR (Fourier Transform Infrared Spectrometer) spectrometer (μ-FITR, Model 102) manufactured by Designs and Prototypes, Ltd., USA, in Nimoiya's Spectral Lab. of Geological Survey of Japan. The emissivity measurements of the samples of relatively low temperature (typically < 60 degree in Celsius) at the natural surfaces are difficult in achieving high S/N (Signal-Noise) ratio, thus, the measurements were generally made for the polished surfaces of the highly heated samples using

the plate heating pot in the laboratory. This instrument provides spectral coverage from 2 to 16 μm with 6 cm$^{-1}$ spectral resolution (which means band width) [41]. We collected three samples around Laoheishan and Huoshaoshan; the lithologic characters of these samples are listed in Table 3. As shown in Figure 5, the emissivity curve of sample WD-$_{2w}$ is displaying as flat one, which is the weathered surface of lava covered by some dry lichen-a symbiotic complex of algae and fungi [42]. As for the other samples, the high emissivity is located around ASTER band 10, and low emissivity of these lavas located around ASTER band 12 as shown in Figure 5. Moreover, the depth of low emissivity near 9.6 μm for these samples is quite different (such as WD-$_{3c}$ and WD-$_{3w}$), which indicates that there is a spectral difference with increase of weathered degree. So far, we can highlight these lava flows by using Lava Flow Index (LFI) of B10/B12 in order to distinguish whether the lava flows are fresh one or weathered one covered by dry lichen or low vegetation. This is different from using the Mafic Index (MI = B12/B13) to extract lithologic information for mafic-ultramafic rocks, as suggested by Ninomiya et al. [14].

Detailed description of samples WD$_1$, WD$_2$, and WD$_3$ see Table 3.

**Table 3.** Lithologic characters of samples collected from the Wudalianchi volcanoes.

| Sample No. | Lithologic Characters | Location |
|---|---|---|
| WD-$_1$ | fresh surface of black lava | South part of Huoshaoshan |
| WD-$_{2c}$ | fresh cut surface of lava | North part of Laoheishan |
| WD-$_{2w}$ | weathered surface of lava covered by some dry lichen | North part of Laoheishan |
| WD-$_{3c}$ | fresh cut surface of lava | South part of Laoheishan |
| WD-$_{3w}$ | weathered surface of lava | South part of Laoheishan |

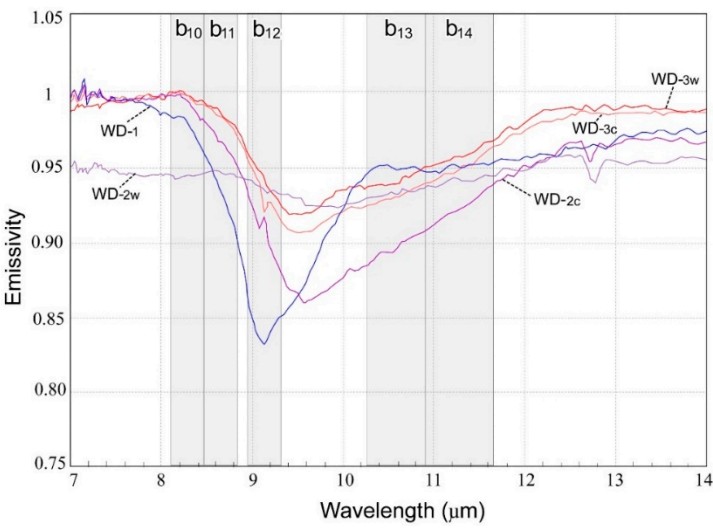

**Figure 5.** Emissivity of typical rock samples collected from Wudalianchi.

Furthermore, band ratios of 2/1 and 4/6 in the VNIR and SWIR regions are used to highlight hematite and altered Al(OH)$_3$ bearing minerals considering that weathered volcanic lavas may contain Hematite and Al(OH)$_3$. Moreover, the band ratios of B2/B1 for displaying as green could highlight the area covered by snow and ice, which is helpful to distinguish the lave flows from snow and ice.

## 4. Results

### 4.1. Topographic and Geomorphologic Features of Quaternary Volcanoes

The ASTER DEMs image shows that the study area is at an elevation range of 175–578 m above sea level as shown by elevation bar in Figure 6a. The 14 major conical volcanic craters are arranged as three sub-parallel zones, extending NE direction, which is similar to the direction of regional fault system [19]

and thus, imply that they are associated with NE-striking normal fault system. The east volcanic zone consists of three shield volcanoes with six conical craters (Molabushan, East Longmenshan, West Longmenshan, Xiaogushan, East Jiaodebushan, West Jiaodebushan), the central zone composed of four shield volcanoes (Weishan, Laoheishan, Huoshaoshan, Wohushan) and two isolated cones (Bijiashan, Yaoquanshan), and the west zone consists of a large shield volcano (North Gelaqiushan, South Gelaqiushan) (Figure 7a). A shield volcano always has a low slope, generally less than 5°, while the diameter of the volcanic pedestal is large, generally more than 3 km. While the cone volcano is mainly composed of basalt and the volcanic cone has a relative height difference of 50 m to 750 m. The shape of cone volcano is just like a cone [43].

A contour image can highlight topographic features of these Quaternary volcanoes (Figures 6b and 7).

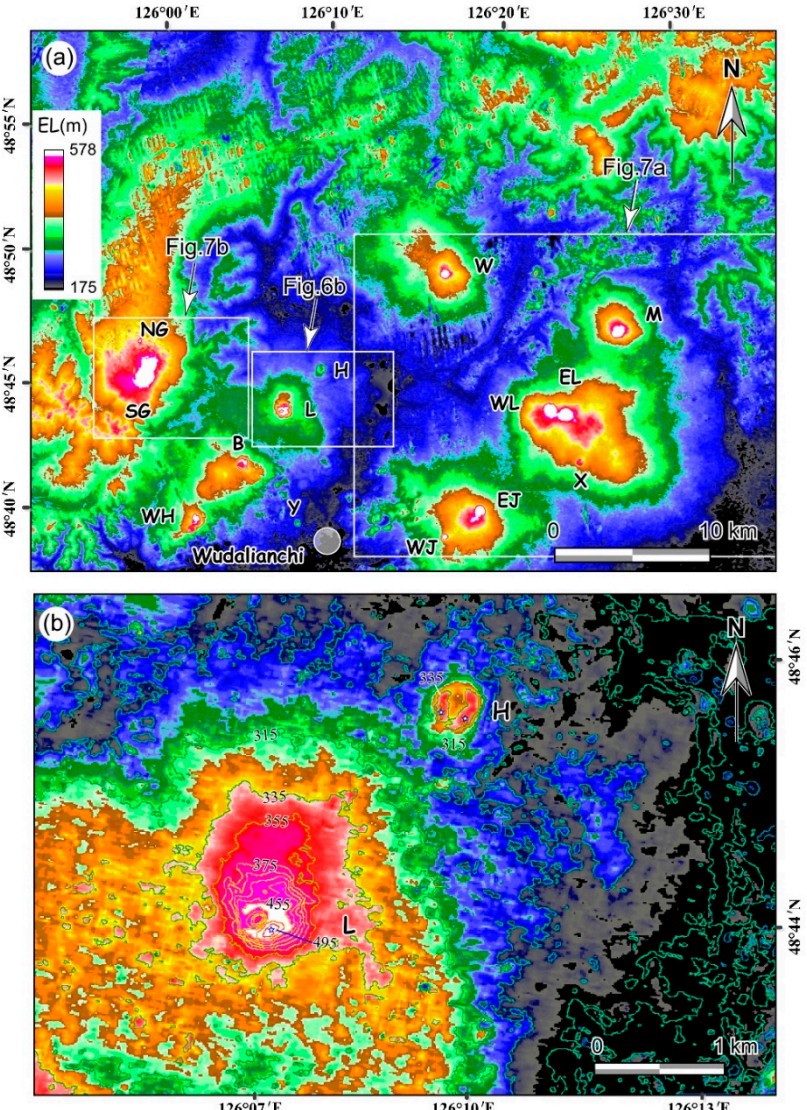

**Figure 6.** (**a**) ASTER DEM image, assigned a step color mode, showing the 14 cone-shaped volcanic craters in the Wudalianchi region. The DEM data obtained on 7 March, 2002. Letters refer to name of volcanic craters are same as Figure 1b. (**b**) Contour image derived from ASTER DEM data showing topographic features of the Laoheishan and Huoshaoshan volcanoes. Contour interval is 20 m, and summits of these craters are indicated by five-shaped stars.

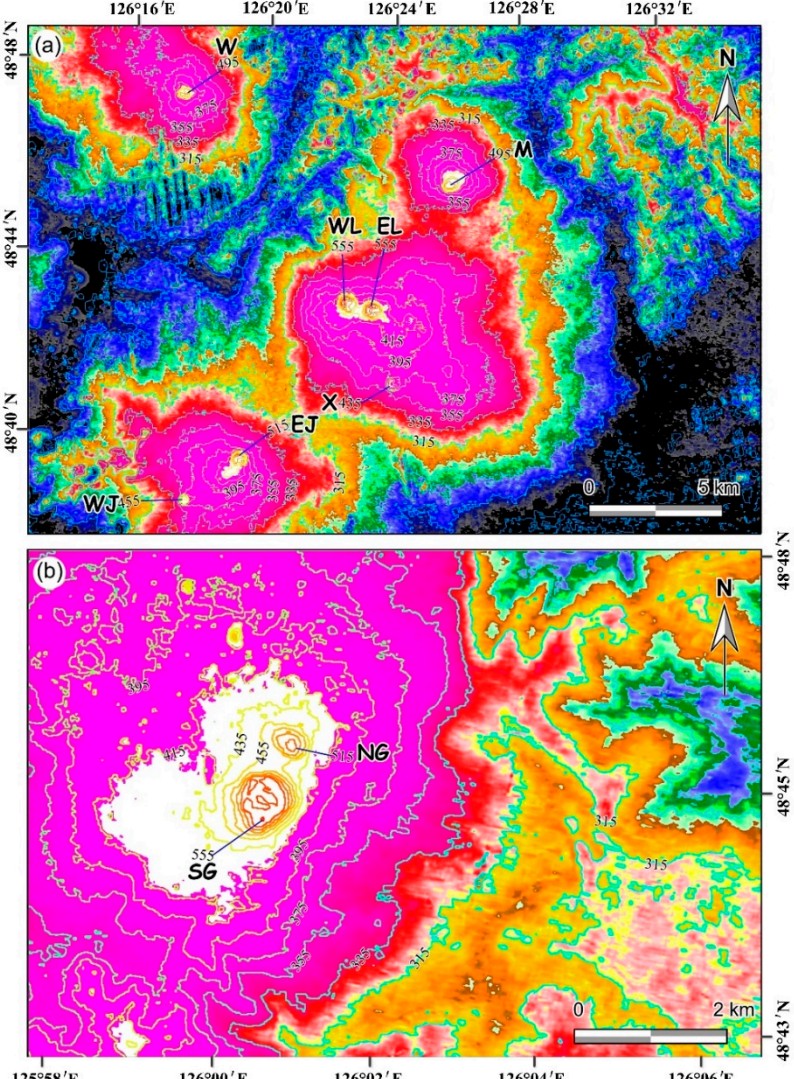

**Figure 7.** Enlarged contour image derived from ASTER data showing the topographic features of volcanic craters in the Wudalianchi volcanic zone. Letters refer to name of volcanic craters are same as Figure 1b, and contour interval is 20 m. (**a**) The east Wudalianchi volcanic zone. (**b**) The west Wudalianchi volcanic zone.

From Figures 6a and 7, Laoheishan volcano is located on southeast and it is about 505 m in height. Meanwhile, there is a deep crater with an elevation of 385 m to west of the summit. However, the Huoshaoshan volcano appears as a northward-facing horseshoe-shaped volcanic landform with two summits about 355 m in height (as indicated by five-shaped stars in Figure 6b). There are four striking shield volcanoes as shown in Figure 7a. The most remarkable shield volcano with a diameter of 2 km is located in middle of image, which consists of three craters (EL, WL, and X in Figure 7a). Two of them (WL and EL) appear as one pair of nearly eastwest extending glasses with a diameter of circa 400 m in the middle eastern part, and the crater of Xiaogushan (X in Figure 7a) in the south flank of EL crater appears as a northeastward-facing horseshoe shape with a diameter of 200 m. The highest elevations of these three carters are 565 m, 570 m, and 445 m (WL, EL, and X, respectively, in Figure 7a). In the north of this large shield volcanic landform, there is a ring shield volcano with a diameter of about 1000 m and summit height of 505 m (M in Figure 7a). The shield volcano in the southwest part of image consists of two craters with summits of 515 m and 465 m, respectively (EJ and WJ in Figure 7a). Another shield volcano has a diameter of 1250 m and a summit of 495 m, appearing in the top-left of image (W in Figure 7a). Figure 7b shows that an NNE striking shield landform with a pair of

cones distributes along the western volcanic zone. The southern zone (SG) exhibits a westward-facing horseshoe shape and the northern zone (NG) exhibits a circular shape with a diameter of 500 m and 220 m, respectively. The summits are about 560 m and 525 m above sea level (SG and NG in Figure 7b) according to the ASTER DEMs contour image.

Three-dimensional (3D) perspective view image can provide an excellent view for geomorphologic features of geologic targets from different view directions [44]. Figure 8 represents an ASTER 3D perspective view image of the Laoheishan and Huoshaoshan volcanoes and adjacent region, which exhibits geomorphologic features of these historic volcanoes clearly (Figure 8).

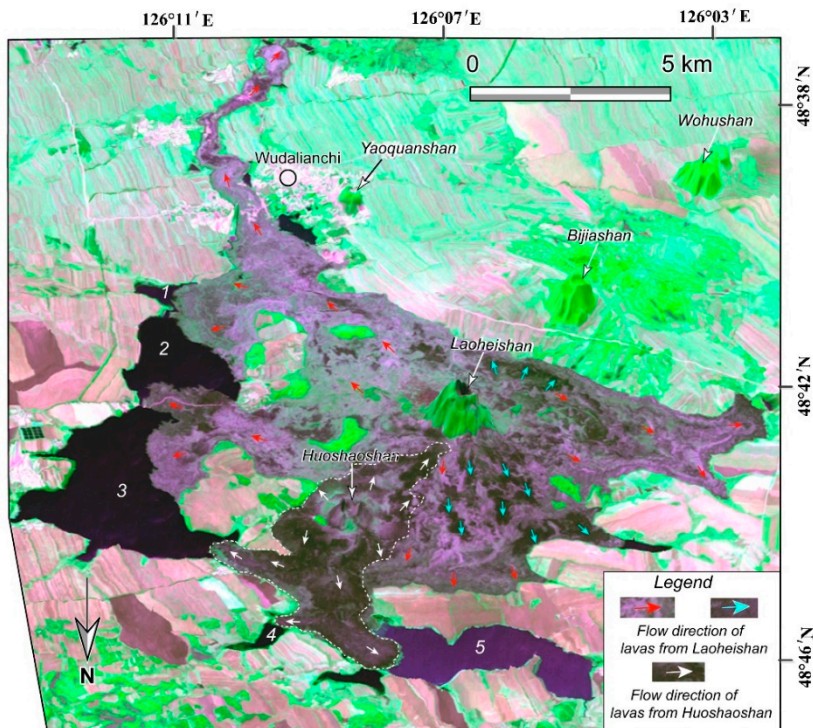

**Figure 8.** ASTER 3D perspective view image (taken on 6 Sepember, 2002, with a three-time vertical exaggeration), showing lava flow and morphology of the Laoheishan and Huoshaoshan volcanoes. Three bands of ASTER VNIR data used to generate false color composite image (bands 2, 3, and 1 assigned in red, green, and blue, respectively). Green and dark blue color patterns represent vegetation and water body, and the five lakes are marked by numbers 1 to 5. Southward-looking view. The red and cyan arrow marks represent two different types of flow directions of lavas from Laoheishan crater and the white arrow marks represent the flow direction of lavas from Huoshaoshan crater.

Laoheishan crater covered with green vegetation exhibits a cone and locates in the central of the Three-dimensional (3D) perspective view image (Figure 8). About 2.5 km northeast of Laoheishan, Huoshaoshan crater shows a much more broken cone with a half-size of Laoheishan crater (Figure 8). Another geomorphologic feature is lava flows from these two historic volcanoes as shown in Figure 8. Based on different color tones and texture characteristics displayed on remote sensing images, different lavas of Laoheishan and Huoshaoshan can be distinguished. Lava flows from Laoheishan crater can be classified into two types: one is the reddish grey lava exhibiting a radial pattern and flowing as far as 15 km (indicated by red arrows in Figure 8), and the other is the darkish grey to black lava distributed around the crater (indicated by blue arrows in Figure 8). Black lava from the Huoshaoshan crater mainly flowed northward and southward (indicated by white arrows in Figure 8). Flow directions of lavas can be distinguished clearly according to the 3D perspective view image combined with the field observation, although over 280 years has passed. The 3D perspective view image also shows that several lakes distribute around the north and east part of these two volcanoes. The Wudalianchi is

named after the Five Large Connected Pools, encircling Laoheishan and Huoshaoshan, as shown in Figure 8. The lakes were formed after the historic eruptions, when molten lava flows, pouring down from the crater, partially blocked the Baihe River.

### 4.2. Lithologic Mapping of Lava Products and Their Relative Age Dating

A false color composite image of B10/ B12, B2/B1, and B4/B6, assigned as R, G, and B, was derived to enhance subtle spectral change of products from lava flows in different stage of eruptions, as shown in Figure 9. Figure 9 shows that the volcanic lava flows have quite different color patterns (purplish red to bright red), which may represent lava flows formed in different eruptive stages as indicated by Kahle et al. [8] for Hawaiin lava flows by using airborne TIMS (Thermal Infrared Multispectral Scanner) images. In the Wudalianchi region, the lava rocks are strong alkaline potassium-rich volcanic rock with an average K (Potassium) content of 5.28% and average $SiO_2$ content of 50.46%. The mineral composition has not big difference. Therefore, we consider that the different eruptive stages are mainly responsible for these color patterns. During the field investigation, we had observed the difference in vegetation coverage, which can also affect these color patterns in the Wudalianchi area.

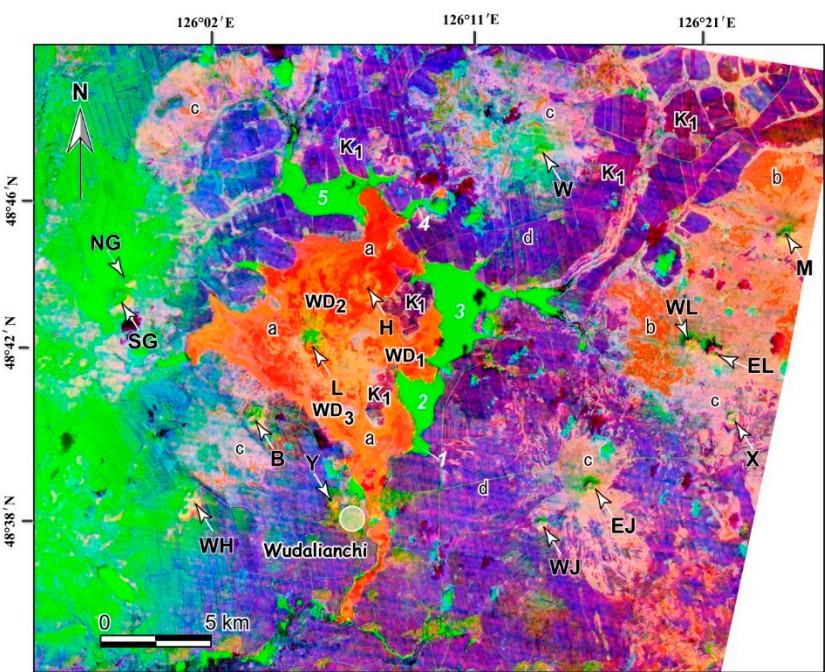

**Figure 9.** ASTER composite image (taken on 7 March, 2002), band ratios of 10/12, 2/1, and 4/6 displayed as red, green, and blue, respectively, showing the lithologic units in the Wudalianchi region. Letters refer to volcanic craters are same as Figure 1b. Letters "a", "b", "c", and "d" refer to relative dating for Quaternary lava flows. "$WD_1$", "$WD_2$", "$WD_3$" refer to the location of samples. "$K_1$" shows the Cretaceous sandstones and argillites. Bright green color patterns represent snow and ice. The five lakes of Wudalianchi are marked by numbers 1 to 5.

According to the results shown in Figure 9, the Wudalianchi Quaternary lava flows can be divided into at least four stages. Compared with the published maps of these flows in [27], the four stages are: (1) historic lava flows from Laoheishan and Huoshaoshan craters showing a bright red pattern (indicated by letter a in Figure 9), which formed by the most recent eruptions in 1719–1721 AD and 1776 AD as recorded by historic documents [17,28]; (2) lava flows distributed around the west flank of West Longmenshan (WL) and north flank of Molabushan showing a reddish pattern, which represent younger volcanoes (indicated by letter b in Figure 9) and belong to the Holocene [27]; (3) lava flows showing a pink pattern in Figure 9 (indicated by letter c), which may represent the product of third stage abruptions, which is the alkaline basalt belonging to the Late Pleistocene [27]; (4) the purplish

blue pattern shows the products from the oldest eruption in the Wudalianchi region (indicated by letter d in Figure 9), which is the alkaline basalt belonging to the Early-middle Pleistocene [27]. In addition, the regions with a purplish red pattern show the Cretaceous sandstones and argillites as observed in the field (indicated by the letter $K_1$ in Figure 9). Our results are consistent with the regional geological map [27], except that they suggested the lava flows among Molabushan (M) belong to the Late Pleistocene, while our results demonstrated that they are at the same stage with the lava flows distributed around west flank of West Longmenshan (WL), which belong to the Holocene.

The enlarged ASTER image (Figure 10) displays more detailed lithologic features of the Laoheishan and Huoshaoshan volcanoes. Although the lava flows around the Laoheishan and Huoshaoshan volcanoes belong to the same stage, their color tones still have subtle differences.

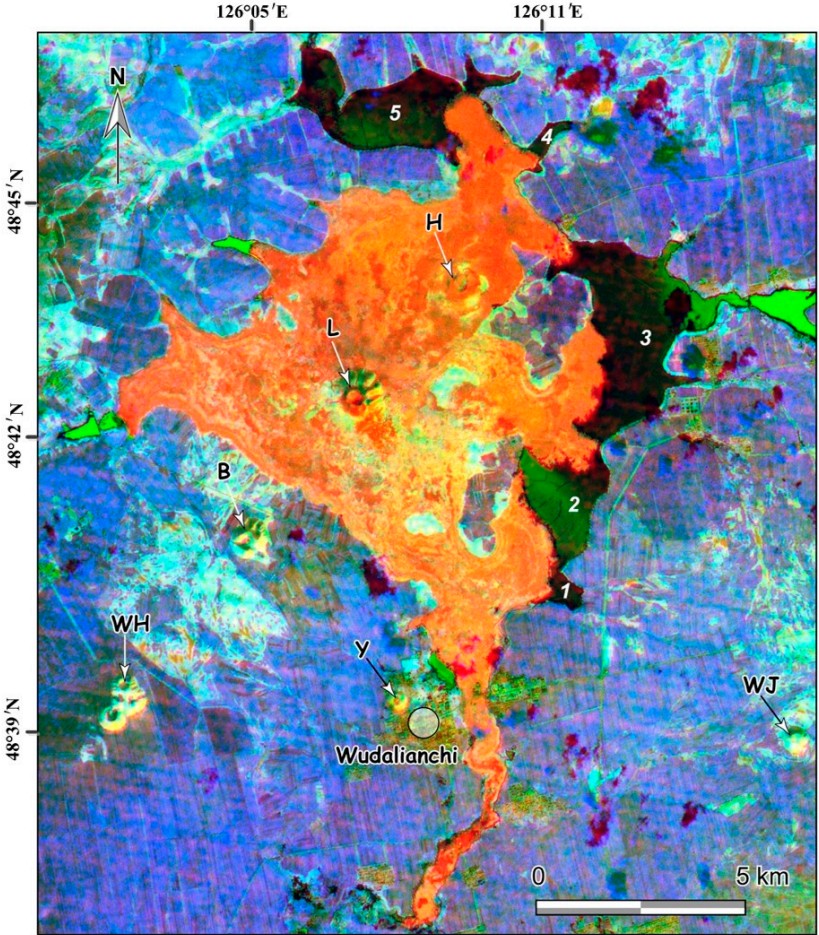

**Figure 10.** Enlarged ASTER composite image (taken on 6 April, 2004), band ratios of 10/12, 2/1, and 4/6 displayed as red, green, and blue, respectively, showing the detailed lithologic units of historical lava flows. Letters refer to volcanic craters are same as Figure 1b. The five lakes of Wudalianchi are marked by numbers 1 to 5.

## 5. Discussion

The quality of the available digital elevation models (DEMs) is crucial for the mapping topographic and geomorphologic features of Quaternary volcanoes. ASTER stereo imaging system can provide high quality DEMs data as shown in Figures 6 and 7. Contour images (Figures 6b and 7) in the Wudalianchi volcanic region demonstrate that ASTER DEMs data have a high vertical accuracy of 20 m without ground control points [12]. It can provide detailed topographic and geomorphologic features of the volcanic landforms with a low relief contrast (50–160 m) in the Wudalianchi region. The resolution of ASTER VNIR data up to 15 m is perfectly suited for large overviews of volcanoes. 3D perspective view

image generated from ASTER VNIR and DEMs data provides excellent views for geomorphologic feature of the volcanoes (Figure 8).

According to the ASTER DEMs (Figures 2, 6 and 7), the most of volcanoes are distributed along the NE-trending faults. We inferred that these conjugate faults might provide the pathway for magma migration during the volcanic eruption [19].

As shown in Figure 9, lava flows from different eruption stages show different color patterns in ASTER multispectral ratio images. Interpretations of these images suggest that at least four stages of volcanic eruptions occurred during the Quaternary. This interpretation for relative dating has a good agreement with geologic mapping given by [27], except for the different judgments of what stage that the lava flows around Molabushan (M) should belong to. However, early studies suggested that there were three stages of lava flows: Early-middle Pleistocene (circa $1.33 \pm 0.08$ to $0.8 \pm 0.02$ Ma (million years), Late Pleistocene (circa 0.63–0.3 Ma), and historic periods [18,26]. Therefore, it is necessary to remeasure age for the lava flows in the Wudalianchi volcanic results revealed by this study.

Another point that needs to be addressed is the subtle differences of lava flows' color tones among the Laoheishan and Huoshaoshan volcanoes. Concerning the different color pattern in ASTER image of these lava flows represent the TIR spectral differences, we suggest that at least two different aspects may have been responsible for subtle spectral differences:

(1) The effects of terrestrial weathering of lava flows. There is a spectral difference with increase of weathered degree as shown in Figure 5 and Table 3. Similar results also revealed that the weathering of Hawaiian basalts has caused their spectral changes [8,9].

(2) The presence of vegetation on lava flows. Spectral measurement shows that the value of emissivity in band 12 increases comparing weathered surface covered by the lichen (WD-$_{2w}$) with cut surface of same sample (WD-$_{2c}$ in Figure 5). Field investigation shows that some lava flows are indeed covered by dry lichen or low brushes although vegetation is sparsely in the Wudalianchi volcanic region (Figure 11).

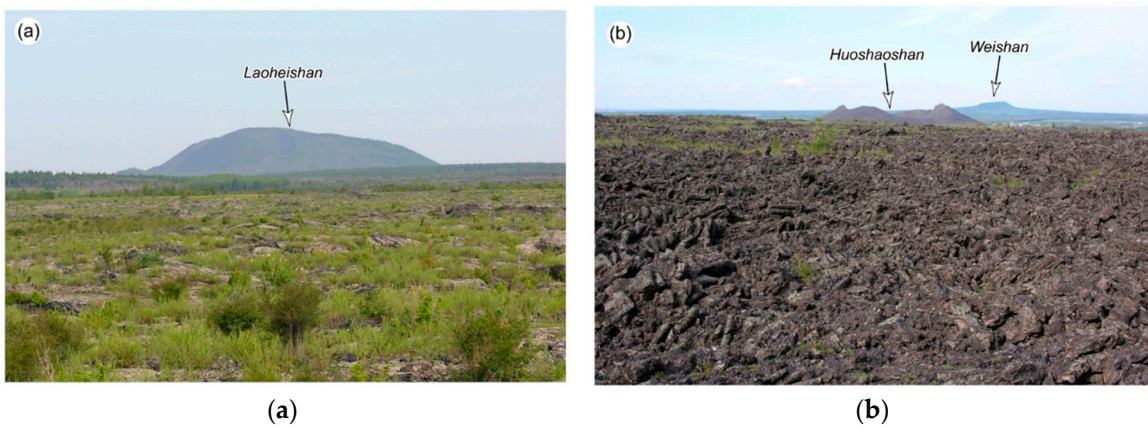

**Figure 11.** Field photographs (taken on August 2005). (**a**) The crater of the Laoheishan and lava flows from 1719–1721 AD eruptions. Northward-looking. (**b**) The lava flows of 1719–1721 AD eruptions from the Huoshaoshan volcano. The crater is the Huoshaoshan in the north part. Northeast-looking. Note lava flows covered by sparse vegetation.

## 6. Conclusions

ASTER DEMs data can provide detailed topographic features for describing volcanic landforms. The resolution of ASTER VNIR data up to 15 m is perfectly suited for large overviews of volcanoes. In this paper, through a series of processing of ASTER images and field investigation, it is implied that the Wudalianchi volcanoes are extending in a NE direction, which is likely influenced by the regional conjugate faults.

ASTER 3D perspective view images can provide excellent view for geomorphologic features of volcanoes, and thus, ASTER stereo imaging gives geoscientists a comprehensive tool for generating high quality topographic images of Quaternary volcanoes. Among the 14 volcanic craters in Wudalianchi region, the molten lava flows of Laoheishan and Huoshaoshan volcanoes partially blocked the Baihe River, which forms the Five Large Connected Pools, known as the Wudalianchi Lake.

Lithologic mapping indicates that at least four stages of volcanic eruptions are revealed in the Wudalianchi Quaternary volcano cluster. The lava flows from different stages of Quaternary volcanic eruptions are mapped successfully in the Wudalianchi volcanic cluster. These results demonstrate that ASTER multispectral data, particularly, the Lava Flow Index (LFI) (LFI = B10/B12) imagery, can be used to map subtle spectral variations caused by the surface weathering. Mapping of these lava flows on the basis of spectral properties may allow us to discriminate the relative age of the lava units in the sparsely vegetated region with arid and semi-arid climate conditions on the earth.

Although this study successfully distinguished four different formation stages of Wudalianchi volcanic lavas, it is still necessary to remeasure age and mineral component for the lava flows in the future research, to figure out what the major reason for these lava flows is with different color patterns in the ASTER image, which represent spectral differences.

In general, ASTER covers a wide spectral region with 14 bands from visible to thermal infrared with high spatial, spectral, and radiometric resolution. Therefore, ASTER can provide an effective approach for mapping the products from late Quaternary volcanoes.

**Author Contributions:** Conceptualization, B.F. and Y.N.; methodology, B.F. and Y.N.; software, B.F.; validation, B.F. and Y.N.; formal analysis, H.F. and B.F.; investigation, B.F. and Y.N.; resources, B.F., Y.N., H.F. and P.S.; data curation, B.F., Y.N., H.F. and P.S.; writing—original draft preparation, H.F. and B.F.; writing—review and editing, H.F. and B.F.; visualization, H.F. and B.F.; supervision, B.F.; project administration, B.F.; funding acquisition, B.F.

**Funding:** This work was supported by the Strategic Priority Research Program of Chinese Academy of Sciences (XDA 20070202) and the National Natural Science Foundation of China (No. 41761144071).

**Conflicts of Interest:** The authors declare no conflict of interest.

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
