# Peer review of "New Insights of Geomorphologic and Lithologic Features on Wudalianchi Volcanoes in the Northeastern China from the ASTER Multispectral Data"

_remotesensing, doi:10.3390/rs11222663_

Round 1
Reviewer 1 Report
Dear authors,
All suggestions and comments have been taken into account. By my side the manuscript could be accepted for publishing in Remote Sensing.
Good job,
Regards,
Author Response
Dear Reviewer,
Thank you so much for your efforts on our manuscript entitled “New insights of geomorphologic and lithologic features on Wudalianchi volcanoes in the Northeastern China from the ASTER multispectral data”, which submitted to the Special Issue "ASTER 20th Anniversary" of Remote Sensing. Your constructive suggestions and comments have helped us improve the content and many key points of the text, tables and figures. And we have learned much from it.
We would like to thank you again for the precious comments from you.
Best regards,
Yours sincerely,
Fu
Corresponding Author:
Prof. Bihong Fu,
Institute of Aerospace Information Research Institute, Chinese Academy of Sciences
No. 9 Dengzhuang South Road, Haidian District, Beijing 100094, P.R. China
Phone: +86-010-8217-8096
Fax: +86-10-8202-4251
E-mail: [email protected]
Reviewer 2 Report
New Insights of Geomorphologic and Lithologic Features on Wudalianchi Volcanoes from the ASTER Multispectral Data
by Fu et al.,
General comments:
This paper illustrates how remote sensing can be used for lithologic mapping. In this work, the authors show the possibility to derive lithological map and distinguish lava flow unit based on ASTER dataet. This study is quite interesting; however, there are some comments especially in methods, results, discussion and conclusion that should be addressed by the authors:
Major comments:
For your data selection you have March 07 2002, Sep 06 2002 and April 06 2004. Why did you apply lithologic maps only for March 2002 and April 2004 not in September 2002? The same question for 3D perspective image why did you only apply in September 2002? Is that any consideration with these datasets? You mention in Ln 350 that ´A contour image can highlight topographic features of these Quaternary volcanoes´. Do you think selection of contour interval will affect this topographic features? And why did you select 20 m interval on your contour? Ln 468-469. ´ASTER DEMs data has a high vertical accuracy of 15 m without ground control points´ Any reference for this statement? You mention in Ln 484-493 at least you have 4 stages of volcanic eruption instead of 3 from early study. Lava flows from different eruption stage show different colour in band ratio or spectral, but there is possibility that even lava flow in same stage of eruption have different spectral or color tone (example due to high rate of oxidation, sulfate deposit etc like in Hawaii or Iceland) see (Aufaristama et al. 2019; Moore, et al. 1987). You also can see in you case on Figure 10 for the lava flows around the Laoheishan and Huoshaoshan volcanoes belong to the same stage, their color tones still have subtle differences Ln 494-506. Do you think different lava morphology also affected lava flow spectral? (such as A´a or pahoehoe lava flow.) and what kind of lava morphology in your study did you see in the field? See (Amici, Piscini, & Neri, 2014; Head M, Maclean L, Carn A, 2012; Li, Solana, Canters, Chan, & Kervyn, 2015)
References
Amici, S., Piscini, A., & Neri, M. (2014). Reflectance Spectra Measurements of Mt . Etna : A Comparison with Multispectral / Hyperspectral Satellite. Advances in Remote Sensing, 3(December), 235–245. https://doi.org/http://dx.doi.org/10.4236/ars.2014.34016
Aufaristama, M, Hoskuldsson, A., Ulfarsson, M. O., Jonsdottir, I., & Thordarson, T. (2019). The 2014-2015 lava flow field at Holuhraun, Iceland: Using airborne hyperspectral remote sensing for discriminating the lava surface. Remote Sensing. https://doi.org/10.3390/rs11050476
Head M, Maclean L, Carn A (2012) Mapping lava flows from Nyamuragira volcano (1967–2011) with satellite data and automated classification methods. Geomat, Nat Haz Risk 4:119–144
Li, L., Solana, C., Canters, F., Chan, J., & Kervyn, M. (2015). Impact of environmental factors on the spectral characteristics of lava surfaces: field spectrometry of basaltic lava flows on Tenerife, Canary Islands, Spain. Remote Sensing, 7(12), 16986–17012. https://doi.org/10.3390/rs71215864
Moore, R. B., Clague, D. A., Rubin, M., & Bohrson, W. A. (1987). Volcanism in Hawaii. In U.S. Geological Survey Professional Paper 1350 (p. 557). https://doi.org/10.1016/0003-6870(73)90259-7
Author Response
Dear Reviewer,
Thank you so much for your constructive suggestions and comments on our manuscript entitled “New insights of geomorphologic and lithologic features on Wudalianchi volcanoes in the Northeastern China from the ASTER multispectral data”, which submitted to the Special Issue "ASTER 20th Anniversary" of Remote Sensing. According to your suggestions and comments, we have modified the manuscript carefully as following.
Point 1: For your data selection you have March 07 2002, Sep 06 2002 and April 06 2004. Why did you apply lithologic maps only for March 2002 and April 2004 not in September 2002? The same question for 3D perspective image why did you only apply in September 2002? Is that any consideration with these datasets?
Response 1: We selected the date of March 7, 2002 for Figure 9 in order to eliminate the influence of the vegetation coverage. We used different date (April 06, 2004) instead of March 07, 2002 to produce an enlarged image of lithologic map (Figure 10) in order to eliminate the influence of the snow and ice coverage (bright green color area in Figure 9). As for the 3D perspective image, we selected the data acquired on Sept. 06, 2002 because there is no influence of the snow and ice coverage in the Wudalianchi area in September.
Point 2: You mention in Ln 350 that “A contour image can highlight topographic features of these Quaternary volcanoes”. Do you think selection of contour interval will affect these topographic features? And why did you select 20 m interval on your contour?
Response 2: We used the contour map to highlight the detail elevation information of these Quaternary volcanoes. In our opinion, the selection of contour interval wouldn’t affect the topographic features, but it will affect the displaying effects of elevation information of the volcanoes.
The vertical accuracy of the DEM data generated from the Level-1A data is 20 m with 95% confidence without ground control point (GCP) correction for individual scene. Thus, we have selected 20 m interval on the contour maps.
Point 3: Ln 468-469. “ASTER DEMs data has a high vertical accuracy of 15 m without ground control points” Any reference for this statement?
Response 3: Fujisada et al (2005) indicated that the vertical accuracy of the DEM data generated from the Level-1A data is 20 m with 95% confidence without ground control point (GCP) correction for individual scenes. We modified the sentence and added this reference for this statement in new version of manuscript.
Point 4: You mention in Ln 484-493 at least you have 4 stages of volcanic eruption instead of 3 from early study. Lava flows from different eruption stage show different colour in band ratio or spectral, but there is possibility that even lava flow in same stage of eruption have different spectral or color tone (example due to high rate of oxidation, sulfate deposit etc like in Hawaii or Iceland) see (Aufaristama et al. 2019; Moore, et al. 1987). You also can see in you case on Figure 10 for the lava flows around the Laoheishan and Huoshaoshan volcanoes belong to the same stage, their color tones still have subtle differences.
Response 4: Acturally, the lava flows in same stage of eruption may have different spectral features or color tones. But the lava rocks in the Wudalianchi region are strong alkaline potassium-rich volcanic rock with an average K (Potassium) content of 5.28% and average SiO2 content of 50.46%. The mineral composition has not big difference. There is no high rate of oxidation, sulfate deposit etc like in Hawaii or Iceland. That’s why we consider that the different eruptive stages are responsible for these color patterns, which means at least 4 stages of volcanic eruption instead of 3 in the Wudalianchi region.
Point 5: Ln 494-506. Do you think different lava morphology also affected lava flow spectral? (such as A´a or pahoehoe lava flow.) and what kind of lava morphology in your study did you see in the field? See (Amici, Piscini, & Neri, 2014; Aufaristama, Höskuldsson, Jónsdóttir, & Ólafsdóttir, 2016; Li, Solana, Canters, Chan, & Kervyn, 2015)
Response 5: According to the field investigation and the regional geological map in reference [27], only the lava flows from the most recent eruption volcanoes, such as Laoheishan and Huoshaoshan, are mainly pahoehoe lava flows and only a few of A´a lava flows developed in the Wudalianchi region. Moreover, their color tones are basically consistent. Thus, the different lava morphology could not affect spectral features of lava flow in the Wudalianchi region.
Best regards,
Yours sincerely,
Fu
Corresponding Author:
Prof. Bihong Fu,
Institute of Aerospace Information Research Institute, Chinese Academy of Sciences
No. 9 Dengzhuang South Road, Haidian District, Beijing 100094, P.R. China
Phone: +86-010-8217-8096
Fax: +86-10-8202-4251
E-mail: [email protected]
Reviewer 3 Report
Response to comments and suggestions from Reviewer #4
Point 1: Line 14: It doesn’t “have” the capability to measure in SWIR, it “had” the capability until 2006.
Response 1: We have changed this sentence following your suggestion.
Point 2: Line 26: imaging?
Response 2: We have changed “imaging” to “data”.
Point 3: Line 34-35: remote sensing can provide a certain resolution of spectral, spatial, and temporal coverage based on the type of sensor, but saying it provides spectral, spatial and temporal coverage doesn’t have a clear meaning.
Response 3: Following your suggestion, we have changed “Remote sensing provides spectral, spatial and temporal coverage” into “Remote sensing can provide a certain resolution of spectral, spatial, and temporal coverage based on the type of sensor”.
Point 4: Line 42: TIMs is brought up here, is there a point? Consider removing.
Response 4: We want to explain further the spectral differences of lava flows caused by the weathering can be measured and mapped with multispectral imaging system.
Another Response to Response 4: This is still out of place and isn’t really related to any other points made in the manuscript. If including this information is important to the manuscript, be clear within the manuscript why it is important.
Response: Kahle et al indicated that the different color patterns of volcanic lava flows may represent lava flows formed in different eruptive stages for Hawaiin lava flows by using airborne TIMS (Thermal Infrared Multispectral Scanner) images in [8], which has provided a good example for us to analyze the lava products and their relative age dating in Wudalianchi region.
Right, but it seems as if that point you want to make here is that “Kahle et al indicated that the different color patterns of volcanic lava flows may represent lava flows formed in different eruptive stages for Hawaiin lava flows” using a TIR multi-spectral sensor not that TIMS specifically was used for this. As it is written, it seems as if TIMS specifically is important for this research, but as you later go on the demonstrate, it can be done with other sensors and TIMS is never again mentioned.
Point 5: Line 46-47: Covered, be clear SWIR is no longer functioning, correct throughout
Response 5: Change “covers” to “covered”.
Point 6: Line 52: geologic applications is probably too broad here, the temperature alone from TM and ETM has geologic applications it just isn’t useful for determining composition. Be more specific.
Response 6: We have supplemented the specific applications of ASTER TIR data, such as determining the relative age dating of lavas, on page 2, line 55-56.
Point 7: Line 55-56: “Therefore…. surface.” this sentence is redundant with the previous sentence, combine or remove.
Response 7: This sentence has been removed.
Point 8: Line 57-58: “Especially” is not needed here; you are going straight for discussing emissivity in one sentence to temperature in the next without distinguishing between the two.
Response 8: “Especially” has been deleted.
Point 9: Line 58-61: This sentence needs to be re-written more clearly; now you are discussing the benefit of global coverage without establishing ASTER coverage, consider introducing this earlier in this paragraph
Response 9: We have modified this sentence as “Therefore, ASTER can provide a potential tool for mapping the products from active volcanoes from regional to global scales”.
Point 10: Line 62: name the volcanoes
Response 10: We have marked the name of volcanoes as the Wudalianchi Volcano and Changbaishan Tianchi Volcano in line 69 of page 2.
Point 11: Line 64: what do you mean “it” still lacks? The volcanoes? Research in general?
Response 11: There, we meant that the general research of topographic and geomorphologic features of these Quaternary volcanoes is still lacks. This sentence has been modified.
Point 12: Figure 1: Both figures are lacking a color ramp or explanation of what they variations in grey represent.
Response 12: We have added a color ramp in Figure 1 to explain what they variations in grey represent.
Another Response to Response 12: This should be a specific temperature, not just “high/low”.
Response: We have changed the color ramp in Figure 1 according to your suggestion.
Point 13: Line 74-76: How does this field investigation relate to this study? This just seems tacked on.
Response 13: During the field investigation, we had observed the difference of vegetation coverage in the Wudalianchi volcanoes, which can provide useful information to understand the spectral difference of the lava flows in the Laoheishan and Huoshaoshan lava flows.
Another Response to Response 13: That’s a good reason, but this needs to be explained in the paper, as it is, as previously stated, it seems tacked on.
Response: We added some explanations to the field investigation in line 86-89 of page 4.
Point 14: Paragraph Lines 78-85: Parts of this paragraph are redundant with information in a previous paragraph (lines 62-66). They could probably be combined.
Response 14: We have simplified the sentences in this paragraph.
Point 15: Line 96: this sentence is worded poorly and difficult to understand.
Response 15: We have simplified many paragraphs in section 2, and this sentence has been deleted.
Point 16: Section 2: This section is interesting, but it’s full of superfluous information that doesn’t help to support the methods, results, or conclusions. Consider condensing this into only one paragraph of critical information.
Response 16: In this section, we have simplified many paragraphs in order to understand clearly the geological setting and geodynamics of intraplate volcanoes in the NE China like the Wudalianchi volcanoes.
Point 17: Figure 2: How is this a geodynamic model? What software was used? What inputs? This looks like it was just drawn and is not necessary based on any real information. How is this related to identifying the geomorphologic and lithological features of the volcanoes? Consider removing.
Response 17: Current Figure 3 is a sketch map, which displays the relationship between the active intraplate volcanoes in NE China and the deep subduction of the Pacific slab. This 3D geodynamic model was drawn using Artificial Intelligence software based on analysis of the topographic image as well as geophysical evidence given by the previous studies. Additionally, we have improved the quality of figure according to the new data published by Wei et al (2018).
Another Response to Response 17: This should be explained in the paper, it’s still not entirely clear. You call it both a sketch map and a 3D model, was it generated using AI, or is it based on the results of the program?
Response: This figure is a 3D geodynamic sketch map which was drawn using Artificial Intelligence (AI) software based on analysis of the topographic image as well as geophysical data given by the previous studies. In order to clarify this point, we changed “model” to “sketch map”. Additionally, we explained this in the manuscript on page 5, line 132-136.
Point 18: Line 130: Results
Response 18: Yes, it should be “Results”.
Point 19: Table 1: This table is not needed, the acquisition timing and DEM specification all should have been included in a methods section.
Response 19: We have adjusted the structure of manuscript, and put this table in the methodology section.
Point 20: Line 137: “It’s clear to see” is subjective, consider removing
Response 20: We have corrected it according to reviewer’s suggestion on page 12, line 340.
Point 21: Line 140-142: How does Figure 3 demonstrate that these volcanoes are shield or cone volcanoes? What indicating features are you using?
Response 21: We have supplemented the definition of the shield and cone volcanoes at this part in line 346-349 of page 12.
Point 22: Line 151-170: How were these heights determined? Methods are needed.
Response 22: These heights are determined by the ASTER DEM (digital elevation model) data, which is an entity ground model that represents the ground elevation in the form of an ordered array of numerical values. The contour image was generated automatically from the DEM data derived from ASTER Level-3A data by using ER-Mapper (Earth Resource-Mapper) software, which could show the elevation of the areas.
Point 23: Line 177-179: Sentence unclear, what is being done here?
Response 23: In this part, we superimposed a three bands color composite image on the ASTER DEM data to generate a 3D perspective view image. In this color composite image, the bands 2, 3, and 1 assigned in red, green and blue, respectively. And from the 3D perspective view image, we could observe the geomorphologic features of geologic targets from different view directions.
Point 24: Line 182: remarkable is subjective.
Response 24: We have modified it.
Point 25: Line 183: What do you mean by “damaged”?
Response 25: The “damaged” means the cone of Huoshaoshan is incomplete compared with the Laoheishan crater.
Another Response to Response 25: This is still unclear and has not been addresses in the manuscript.
Response: We used “broken” to substitute “damaged”.
Point 26: Line 184: “spectacular” is subjective, as is “clearly”
Response 26: We have removed “spectacular”.
Point 27: Line 188/189/Figure 5: what distinguishes the red arrow and blue arrow lava flows? Is there a border? Why do you see this difference?
Response 27: We analyzed flow direction of lavas according to the texture characteristics and color tone of the lava flow.
Another Response to Response 27: This does not answer the question and was not addressed in the manuscript.
Response: We distinguished different lava flows based on their different color tones and texture characteristics. Because different types of lavas have different spectral characteristics, so that different color tones and texture characteristics are displaying on remote sensing images. Based on this, the lava flows from the Laoheishan and Huoshaoshan can be distinguished clearly. Lava flows from Laoheishan crater can be classified into two types: lavas in reddish grey exhibit radial pattern and flowed as far as 15 km (indicated by red arrows on new Figure 8), and ones in darkish grey to black colors are just distributed around the Laoheishan crater (indicated by blue arrows on new Figure 8). However, lavas from Huoshaoshan crater in black color mainly flows northward and southward (indicated by white arrows on Figure 8). Flow directions of lavas can be distinguished clearly according to the 3D perspective view image combined with the field observation. We marked an explanation on page 16, line 402-411.
Point 28: Figure 5: What is the vertical exaggeration?
Response 28: We exaggerated topographic features vertically in order to display clearly the morphologic features of these young volcanoes.
Another Response to Response 28: This is an overlain DEM image, the DEM has some vertical exaggeration value, what is it.
Response: In order to display clearly the topographical features of the Wudalianchi volcanic group and determine the flow direction of the lavas, we exaggerated the elevation value of the DEM by 3 times in the vertical direction.
Alright, then you need to include that this figure has a vertical exaggeration of 3 in the figure caption.
Point 29: Line 211-213: What is the significance of this lava flow index? What are you trying to identify/index?
Response 29: We defined this lava flow index in order to identify the spectral subtle variations of these lava flows with different age in the Wudalianchi volcanoes.
Another Response to Response 29: This needs to be addressed in the manuscript, this description still does not clearly define what this lava flow index is or why it’s significant. It this what you’re calling the spectral library? Or the list of sample locations?
Response: This lava flow index is a ratio of ASTER band 10 and band 12 based on analysis of the laboratory emissivity of typical lava rocks. As shown in new Figure 5, the emissivity curve of sample WD-2w is displaying as flat one, which is weathered surface of lava covered by some dry lichen. For the other samples, the high emissivity is located around ASTER band 10, and low emissivity of these lavas located around ASTER band 12. Moreover, the depth of low emissivity near 9.6 µm for these samples is quite different. Thus, we highlight these lava flows by using ASTER ratio technique in order to distinguish whether the lava flows are fresh one or weathered one covered by dry lichen or low vegetation. We have also explained this in the manuscript.
I can see you’ve addressed this, but the way it’s written is still clear. Try, “ So far, we can highlight these lava flows by using lava flow index (LFI) of LFI-B10/B12 in order to distinguish whether the lava flows are fresh one or weathered one covered by dry lichen or low vegetation.
Point 30: Figure 6/Table 2: Is the point of this table just to demonstrate what the average spectra of the lava flow are? If so, is there a need to include specific spectra rather than an averaged spectrum? Also, how were these spectra collected? In what lab? Averaged over how many scans? Are they continuum removed? Where they heated? Please include in methods.
Response 30: Figure 6 (the new Figure 5) shows the laboratory emissivity in different bands of those samples. We use Figure 6 (the new Figure 5) to determine how to perform bands operations to highlight differences of spectral emissivity between different samples in order to better distinguish different types of samples.
In Nimoiya’s Spectral Lab. of Geological Survey of Japan, we measured the laboratory emissivity by using portable FTIR (Fourier Transform Infrared Spectrometer) spectrometer (µ-FITR, Model 102) manufactured by Designs and Prototypes, Ltd., USA. The emissivity measurements of the samples of relatively low temperature (typically < 60 degree in Celsius) at the natural surfaces are difficult in achieving high S/N ratio, so the measurements were generally made for the polished surfaces of the highly heated samples using the plate heating pot in the laboratory.
Another Response to Response 30: This needs to be included in the methods section of the paper and expanded to address the original questions.
Response: According to your suggestion, we added this part in the methods section of the manuscript.
Point 31: Table 2: Which lava flow were they associated with? Can you add these points on a map? Can you be more descriptive in the lithology? Did you actually scan the region with the lichen in 2W? If so, what is the purpose of this?
Response 31: We marked the sampling locations on new Figure 9, which are associated with the lava flows of Laoheishan and Huoshaoshan volcanoes. We measured the weathered surface of sample WD-2w partially covered by the dry lichen as shown in new Figure 5. The emissivity curve of sample WD-2w is displaying as flat one compared with that of fresh surface.
Another Response to Response 31: Point the reader to figure 8 in the description for table 2, the lichen explanation needs to be included in methods.
Response: We have given a brief introduction of the samples in the methods section as following: “We collected 3 samples around Laoheishan and Huoshaoshan, the lithologic characters of these samples are listed in Table 3 and the location of these samples marked in Figure 9” in line 285-287 of page 9.
In addition, the lichen explanation was also included in this part, following your suggestion.
Point 32: Line 225-226: Is different eruptive stages the only possibility for these color patterns? Expand on what these different colors mean? Do they relate to changes in roughness? Composition? Vescularity? Alteration? What is the significance of figures 7 and 8 beyond just having different colors for different flows?
Response 32: In the Wudalianchi region, the lava rocks are the high-K (Potassium) basalt content with an average of 5.28% and average SiO2 (Silicondioxide) content of 50.46%. The mineral composition has not big difference. Therefore, we consider that both the facies and weathering differences of volcanic rocks in the different eruptive stages are responsible for these color patterns. During the field investigation, we had observed the difference in vegetation coverage, which can also affect these color patterns in the Wudalianchi area.
Another Response to Response 32: Not addressed in manuscript.
Response: According to your suggestion, we modified the manuscript by adding these explanations on page 16, line 427-430.
Point 33: Lines 230-236: How do you make the connection of the different color flows to the different timing of flows? Is this compared against established maps of these flows? If so, what is the significance of these findings?
Response 33: We have compared this against the established maps in Reference [27]. Our interpretation for relative dating has a good agreement with geologic mapping given by [27], but the spatial distribution of these lava flows in different stages is different with that revealed by the previous study. Therefore, it is necessary to re-measure age for the lava flows in the Wudalianchi volcanic results revealed by this study.
Another Response to Response 33: Not explained clearly.
Response: We consider that the different eruptive stages are mainly responsible for these color pattern (refer the Response 32). According to the results shown in Figure 8 (the new Figure 9), the Wudalianchi Quaternary lava flows can be divided into at least four stages. They are similar with the published maps of these flows in [27], the four stages are respectively: (1) historic lava flows from Laoheishan and Huoshaoshan craters showing bright red pattern (indicated by letter a on new Figure 9), which formed by the most recent eruptions in 1719-1721 AD and 1776 AD as recorded by historic documents [17,28]; (2) lava flows distributed around west flank of West Longmenshan (WL) and north flank of Molabushan showing reddish pattern, which represent youngest volcanoes (indicated by letter b on new Figure 9) and belong to the Holocene [27]; (3) lava flows showing pink pattern on new Figure 9 (indicated by letter c) may represent product of third stage abruptions, which are alkaline basalt belong to the Late Pleistocene [27]. Finally, purplish red to purplish blue pattern should be products from the oldest eruption in the Wudalianchi region (indicated by letter d on new Figure 9), which are alkaline basalt belong to the Early-middle Pleistocene [27].
Point 34: Figure 7 and 8: What is the significance of the numbers of this figure? It’s not clear how this relative dating was made, please expand.
Response 34: The numbers 1-5 in new Figure 9 and 10 refer to the five lakes that make up the Wudalianchi Pool. And letters a-d represent the relative age dating of lava rocks according to the color tones of the lava flows on new Figure 9.
Another Response to Response 34: This does not explain how color tone relates to age.
Response: As shown in new Figure 5, the high emissivity is located around ASTER band 10, and low emissivity of these lavas located around ASTER band 12. Moreover, there is a spectral difference with increase of weathered degree. The emissivity of these lava flows with fresh surface is lower than those lava flows with weathered surface for ASTER band 12, which means the color tone of lava flows is related to their relative age dating.
Good point, point you never mention in the paper that the 10/12 ratio can be used for anything other than identifying vegetation free lava. If you’re also using it to determine the degree of weather which you are using to relate to the relative age of the lava flows, you should include this somewhere in the methods. It would also be useful if you wen into a little more detail about why you are also looking for Hematite and AL(OH)3 to help identify these flows. If you are just using the 10/12 ratio, then what is the purpose of including these other ratios?
Point 35: Line 249: Are you implying that this is the first case of contour mapping in this region? What about STRM, or ground based surveys?
Response 35: We did not mention that this research is the first case of contour mapping in this region. The resolution of ASTER Level-3A DEM is 15m, while the resolution of SRTM is 30m or 90m. In this paper, we try to demonstrate that the high quality of ASTER DEM data, which can almost provide global coverage on earth, is suitable for documenting the topographic and geomorphologic features of volcanoes.
Point 36: Line 253: Earlier you state an accuracy of 20m, is it 15 or 20? Where are you getting these numbers from?
Response 36: We are so sorry to make a mistake in the Table 2. The DEM data we used was derived from the ASTER Level-3A product, with accuracy of 15m. We have corrected it.
Point 37: Line 260: If these flows are already been mapped going back as early as 1979, how is this study novel? How does the spatial distribution different? What is the previous study you are referring to?
Response 37: The map in 1979 suggested the lava flows around Molabushan (M) belong to the Late Pleistocene, while our results demonstrated that they are at the same stage with the lava flows distributed around west flank of West Longmenshan (WL), which belong to the Holocene.
Another Response to Response 37: If this is your primary discovery, please expand on: “Our results basically consistent with the map in [27], except that they did not distinguish between the d and e phases [27].”
Response: After careful analysis and reading of relevant literature, we have modified the Results and Discussion section (refer the Response 33). Our results are basically consistent with the map in [27], except that they suggested the lava flows around Molabushan (M) belong to the Late Pleistocene, while our results demonstrated that they are at the same stage with the lava flows distributed around west flank of West Longmenshan (WL), which belong to the Holocene. We explained this point on page 17, line 439-455.
The color of this flow does match the WL flow, but M is also covered by vegetation. However, SWIR is very sensitive to vegetation. Do you have some evidence that this color is directly related to the flow and just that similar types of vegetation are growing of these two flows? If you are using some method to solve for this, it should be included in the methods.
Point 38: Line 269-276: There are valid questions. Consider attempting to answer these based on the data you have available.
Response 38: We have modified this paragraph clearly as shown in line 495-499 of page 19.
Point 39: Line 283: The DEMS can generate topographic features?
Response 39: We have changed “generate” into “provide”.
Author Response
Dear Reviewer,
Thank you again for your constructive suggestions and comments on our manuscript entitled “New insights of geomorphologic and lithologic features on Wudalianchi volcanoes in the Northeastern China from the ASTER multispectral data”, which submitted to the Special Issue "ASTER 20th Anniversary" of Remote Sensing. Following your suggestions and comments, we have modified the manuscript carefully as shown as follows.
Point 1: Line 14: It doesn’t “have” the capability to measure in SWIR, it “had” the capability until 2006.
Response 1: We have modified this sentence following your suggestion.
Point 2: Line 26: imaging?
Response 2: We have changed “imaging” to “data”.
Point 3: Line 34-35: remote sensing can provide a certain resolution of spectral, spatial, and temporal coverage based on the type of sensor, but saying it provides spectral, spatial and temporal coverage doesn’t have a clear meaning.
Response 3: Following your suggestion, we have changed “Remote sensing provides spectral, spatial and temporal coverage” into “Remote sensing can provide a certain resolution of spectral, spatial, and temporal coverage based on the type of sensor”.
Point 4: Line 42: TIMs is brought up here, is there a point? Consider removing.
Response 4: We intend to explain further the spectral differences of lava flows caused by the weathering can be measured and mapped with multispectral imaging system.
Another Response to Response 4: This is still out of place and isn’t really related to any other points made in the manuscript. If including this information is important to the manuscript, be clear within the manuscript why it is important.
Response: Kahle et al indicated that the different color patterns of volcanic lava flows may represent lava flows formed in different eruptive stages for Hawaiin lava flows by using airborne TIMS (Thermal Infrared Multispectral Scanner) images in [8], which has provided a good example for us to analyze the lava products and their relative age dating in Wudalianchi region.
Another Response to Response 4: Right, but it seems as if that point you want to make here is that “Kahle et al indicated that the different color patterns of volcanic lava flows may represent lava flows formed in different eruptive stages for Hawaiin lava flows” using a TIR multi-spectral sensor not that TIMS specifically was used for this. As it is written, it seems as if TIMS specifically is important for this research, but as you later go on the demonstrate, it can be done with other sensors and TIMS is never again mentioned.
Response: We removed “such as NASA’s airborne Thermal Infrared Mulstispectral Scanner (TIMS)” in the new manuscript.
Point 5: Line 46-47: Covered, be clear SWIR is no longer functioning, correct throughout
Response 5: Change “covers” to “covered”.
Point 6: Line 52: geologic applications is probably too broad here, the temperature alone from TM and ETM has geologic applications it just isn’t useful for determining composition. Be more specific.
Response 6: We have added the specific applications of ASTER TIR data, such as determining the relative age dating of lavas, on page 2, line 56-57.
Point 7: Line 55-56: “Therefore…. surface.” this sentence is redundant with the previous sentence, combine or remove.
Response 7: This sentence has been removed in the new manuscript.
Point 8: Line 57-58: “Especially” is not needed here; you are going straight for discussing emissivity in one sentence to temperature in the next without distinguishing between the two.
Response 8: “Especially” has been deleted.
Point 9: Line 58-61: This sentence needs to be re-written more clearly; now you are discussing the benefit of global coverage without establishing ASTER coverage, consider introducing this earlier in this paragraph
Response 9: We have modified this sentence as “Therefore, ASTER can provide a potential tool for mapping the products from active volcanoes from regional to global scales”.
Point 10: Line 62: name the volcanoes
Response 10: We have marked the name of volcanoes as the Wudalianchi Volcano and Changbaishan Tianchi Volcano in line 66 of page 2.
Point 11: Line 64: what do you mean “it” still lacks? The volcanoes? Research in general?
Response 11: There, we meant that the general research of topographic and geomorphologic features of these Quaternary volcanoes is still lacks. We have modified this sentence in the new manuscript.
Point 12: Figure 1: Both figures are lacking a color ramp or explanation of what they variations in grey represent.
Response 12: We have added a color ramp in Figure 1 to explain what they variations in grey represent.
Another Response to Response 12: This should be a specific temperature, not just “high/low”.
Response: We have changed the color ramp in Figure 1 according to your suggestion.
Point 13: Line 74-76: How does this field investigation relate to this study? This just seems tacked on.
Response 13: During the field investigation, we had observed the difference of vegetation coverage in the Wudalianchi volcanoes, which can provide useful information to understand the spectral difference of the lava flows in the Laoheishan and Huoshaoshan lava flows.
Another Response to Response 13: That’s a good reason, but this needs to be explained in the paper, as it is, as previously stated, it seems tacked on.
Response: We added some explanations to the field investigation in line 83-86 of page 4.
Point 14: Paragraph Lines 78-85: Parts of this paragraph are redundant with information in a previous paragraph (lines 62-66). They could probably be combined.
Response 14: We have simplified the sentences in this paragraph.
Point 15: Line 96: this sentence is worded poorly and difficult to understand.
Response 15: We have simplified many paragraphs in section 2, and this sentence has been deleted.
Point 16: Section 2: This section is interesting, but it’s full of superfluous information that doesn’t help to support the methods, results, or conclusions. Consider condensing this into only one paragraph of critical information.
Response 16: In this section, we have simplified many paragraphs in order to understand clearly the geological setting and geodynamics of intraplate volcanoes in the NE China like the Wudalianchi volcanoes.
Point 17: Figure 2: How is this a geodynamic model? What software was used? What inputs? This looks like it was just drawn and is not necessary based on any real information. How is this related to identifying the geomorphologic and lithological features of the volcanoes? Consider removing.
Response 17: Current Figure 3 is a sketch map, which displays the relationship between the active intraplate volcanoes in NE China and the deep subduction of the Pacific slab. This 3D geodynamic model was drawn using Artificial Intelligence software based on analysis of the topographic image as well as geophysical evidence given by the previous studies. Additionally, we have improved the quality of figure according to the new data published by Wei et al (2018).
Another Response to Response 17: This should be explained in the paper, it’s still not entirely clear. You call it both a sketch map and a 3D model, was it generated using AI, or is it based on the results of the program?
Response: This figure is a 3D geodynamic sketch map which was drawn using Artificial Intelligence (AI) software based on analysis of the topographic image as well as geophysical data given by the previous studies. In order to clarify this point, we changed “model” to “sketch map”. Additionally, we explained this in the manuscript on page 5, line 123-126.
Point 18: Line 130: Results
Response 18: Yes, it should be “Results”.
Point 19: Table 1: This table is not needed, the acquisition timing and DEM specification all should have been included in a methods section.
Response 19: We have adjusted the structure of manuscript, and put this table in the methodology section.
Point 20: Line 137: “It’s clear to see” is subjective, consider removing
Response 20: We have corrected it according to reviewer’s suggestion on page 9, line 199.
Point 21: Line 140-142: How does Figure 3 demonstrate that these volcanoes are shield or cone volcanoes? What indicating features are you using?
Response 21: We have added the definition of the shield and cone volcanoes at this part in line 204-207 of page 9.
Point 22: Line 151-170: How were these heights determined? Methods are needed.
Response 22: These heights are determined by the ASTER DEM (digital elevation model) data, which is an entity ground model that represents the ground elevation in the form of an ordered array of numerical values. The contour image was generated automatically from the DEM data which was derived from ASTER Level-1A data by using ER-Mapper (Earth Resource-Mapper) software, which can show the elevation of the areas.
Point 23: Line 177-179: Sentence unclear, what is being done here?
Response 23: In this part, we superimposed a three bands color composite image on the ASTER DEM data to generate a 3D perspective view image. In this color composite image, the bands 2, 3, and 1 assigned in red, green and blue, respectively. And from the 3D perspective view image, we could observe the geomorphologic features of geologic targets from different view direction.
Point 24: Line 182: remarkable is subjective.
Response 24: We have modified it.
Point 25: Line 183: What do you mean by “damaged”?
Response 25: The “damaged” means the cone of Huoshaoshan is incomplete compared with the Laoheishan crater.
Another Response to Response 25: This is still unclear and has not been addresses in the manuscript.
Response: We used “broken” to replace “damaged”.
Point 26: Line 184: “spectacular” is subjective, as is “clearly”
Response 26: We have removed “spectacular”.
Point 27: Line 188/189/Figure 5: what distinguishes the red arrow and blue arrow lava flows? Is there a border? Why do you see this difference?
Response 27: We analyzed flow direction of lavas according to the texture characteristics and color tone of the lava flow.
Another Response to Response 27: This does not answer the question and was not addressed in the manuscript.
Response: Based on different color tones and texture characteristics displayed on remote sensing images, different lavas of Laoheishan and Huoshaoshan can be distinguished. Lava flows from Laoheishan crater can be classified into two types: one is the reddish grey lavas exhibited radial pattern and flowed as far as 15 km (indicated by red arrows on Figure 8), and another is the lavas in darkish grey to black just distributed around the crater (indicated by blue arrows on Figure 8). Lavas from Huoshaoshan crater in black color mainly flowed northward and southward (indicated by white arrows on Figure 8). Flow directions of lavas can be distinguished clearly according to the 3D perspective view image combined with the field observation. We marked an explanation on page 13, line 258-266.
Point 28: Figure 5: What is the vertical exaggeration?
Response 28: We exaggerated topographic features vertically in order to display clearly the morphologic features of these young volcanoes.
Another Response to Response 28: This is an overlain DEM image, the DEM has some vertical exaggeration value, what is it.
Response: In order to display clearly the topographical features of the Wudalianchi volcanic group and determine the flow direction of the lavas, we exaggerated the elevation value of the DEM by 3 times in the vertical direction.
Another Response to Response 28: Alright, then you need to include that this figure has a vertical exaggeration of 3 in the figure caption.
Response: We have added this information in the caption of Figure 8 as “ASTER 3D perspective view image (taken on Sep. 06, 2002, with a vertical exaggeration of 3 times) showing lava flow and morphology of the Laoheishan and Huoshaoshan volcanoes”.
Point 29: Line 211-213: What is the significance of this lava flow index? What are you trying to identify/index?
Response 29: We defined this lava flow index in order to identify the spectral subtle variations of these lava flows with different age in the Wudalianchi volcanoes.
Another Response to Response 29: This needs to be addressed in the manuscript, this description still does not clearly define what this lava flow index is or why it’s significant. It this what you’re calling the spectral library? Or the list of sample locations?
Response: This lava flow index is a ratio of ASTER band 10 and band 12 based on analysis of the laboratory emissivity of typical lava rocks. As shown in new Figure 5, the emissivity curve of sample WD-2w is displaying as flat one, which is weathered surface of lava covered by some dry lichen. For the other samples, the high emissivity is located around ASTER band 10, and low emissivity of these lavas located around ASTER band 12. Moreover, the depth of low emissivity near 9.6 µm for these samples is quite different. Thus, we highlight these lava flows by using ASTER ratio technique in order to distinguish whether the lava flows are fresh one or weathered one covered by dry lichen or low vegetation. We have also explained this in the manuscript.
Another Response to Response 29: I can see you’ve addressed this, but the way it’s written is still clear. Try, “So far, we can highlight these lava flows by using lava flow index (LFI) of LFI-B10/B12 in order to distinguish whether the lava flows are fresh one or weathered one covered by dry lichen or low vegetation.
Response: We have modified this into “So far, we can highlight these lava flows by using Lava Flow Index (LFI) of B10/B12 in order to distinguish whether the lava flows are fresh one or weathered one covered by dry lichen or low vegetation.” according to your suggestion.
Point 30: Figure 6/Table 2: Is the point of this table just to demonstrate what the average spectra of the lava flow are? If so, is there a need to include specific spectra rather than an averaged spectrum? Also, how were these spectra collected? In what lab? Averaged over how many scans? Are they continuum removed? Where they heated? Please include in methods.
Response 30: Figure 6 (the new Figure 5) shows the laboratory emissivity in different bands of those samples. We use Figure 6 (the new Figure 5) to determine how to perform bands operations to highlight differences of spectral emissivity between different samples in order to better distinguish different types of samples.
In Nimoiya’s Spectral Lab. of Geological Survey of Japan, we measured the laboratory emissivity by using portable FTIR (Fourier Transform Infrared Spectrometer) spectrometer (µ-FITR, Model 102) manufactured by Designs and Prototypes, Ltd., USA. The emissivity measurements of the samples of relatively low temperature (typically < 60 degree in Celsius) at the natural surfaces are difficult in achieving high S/N ratio, so the measurements were generally made for the polished surfaces of the highly heated samples using the plate heating pot in the laboratory.
Another Response to Response 30: This needs to be included in the methods section of the paper and expanded to address the original questions.
Response: According to your suggestion, we added this part in the methods section of the manuscript.
Point 31: Table 2: Which lava flow were they associated with? Can you add these points on a map? Can you be more descriptive in the lithology? Did you actually scan the region with the lichen in 2W? If so, what is the purpose of this?
Response 31: We marked the sampling locations on new Figure 9, which are associated with the lava flows of Laoheishan and Huoshaoshan volcanoes. We measured the weathered surface of sample WD-2w partially covered by the dry lichen as shown in new Figure 5. The emissivity curve of sample WD-2w is displaying as flat one compared with that of fresh surface.
Another Response to Response 31: Point the reader to figure 8 in the description for table 2, the lichen explanation needs to be included in methods.
Response: We have given a brief introduction of the samples in the methods section as following: “We collected 3 samples around Laoheishan and Huoshaoshan, the lithologic characters of these samples are listed in Table 3 and the location of these samples marked in Figure 9” in line 171-173 of page 7.
In addition, the lichen explanation was also included in this part, following your suggestion.
Point 32: Line 225-226: Is different eruptive stages the only possibility for these color patterns? Expand on what these different colors mean? Do they relate to changes in roughness? Composition? Vescularity? Alteration? What is the significance of figures 7 and 8 beyond just having different colors for different flows?
Response 32: In the Wudalianchi region, the lava rocks are strong alkaline potassium-rich volcanic rock with an average K (Potassium) content of 5.28% and average SiO2 content of 50.46%. The mineral composition has not big difference. Therefore, we consider that both the facies and weathering differences of volcanic rocks in the different eruptive stages are responsible for these color patterns. During the field investigation, we had observed the difference in vegetation coverage, which can also affect these color patterns in the Wudalianchi area.
Another Response to Response 32: Not addressed in manuscript.
Response: According to your suggestion, we modified the manuscript by adding these explanations on page 13, line 277-283.
Point 33: Lines 230-236: How do you make the connection of the different color flows to the different timing of flows? Is this compared against established maps of these flows? If so, what is the significance of these findings?
Response 33: We have compared this against the established maps in Reference [27]. Our interpretation for relative dating has a good agreement with geologic mapping given by [27], but the spatial distribution of these lava flows in different stages is different with that revealed by the previous study. Therefore, it is necessary to re-measure age for the lava flows in the Wudalianchi volcanic results revealed by this study.
Another Response to Response 33: Not explained clearly.
Response: We consider that the different eruptive stages are mainly responsible for these color patterns (refer the Response 32). According to the results shown in Figure 8 (the new Figure 9), the Wudalianchi Quaternary lava flows can be divided into at least four stages. They are similar with the published maps of these flows in [27], the four stages are respectively: (1) historic lava flows from Laoheishan and Huoshaoshan craters showing bright red pattern (indicated by letter a on Figure 9), which formed by the most recent eruptions in 1719-1721 AD and 1776 AD as recorded by historic documents [17,28]; (2) lava flows distributed around west flank of West Longmenshan (WL) and north flank of Molabushan showing reddish pattern, which represent younger volcanoes (indicated by letter b on Figure 9) and belong to the Holocene [27]; (3) lava flows showing pink pattern on Figure 9 (indicated by letter c) may represent product of third stage abruptions, which are alkaline basalt belong to the Late Pleistocene [27]. Finally, purplish blue pattern should be products from the oldest eruption in the Wudalianchi region (indicated by letter d on Figure 9), which are alkaline basalt belong to the Early-middle Pleistocene [27]. In addition, the regions with purplish red pattern show the Creatous sandstones and argillites as observed in the field (indicated by ltter K1 on Figure 9).
Point 34: Figure 7 and 8: What is the significance of the numbers of this figure? It’s not clear how this relative dating was made, please expand.
Response 34: The numbers 1-5 in new Figure 9 and 10 refer to the five lakes that make up the Wudalianchi Pool. And letters a-d represent the relative age dating of lava rocks according to the color tones of the lava flows on new Figure 9.
Another Response to Response 34: This does not explain how color tone relates to age.
Response: As shown in new Figure 5, the high emissivity is located around ASTER band 10, and low emissivity of these lavas located around ASTER band 12. Moreover, there is a spectral difference with increase of weathered degree. The emissivity of these lava flows with fresh surface is lower than those lava flows with weathered surface for ASTER band 12, which means the color tone of lava flows is related to their relative age dating.
Another Response to Response 34: Good point, point you never mention in the paper that the 10/12 ratio can be used for anything other than identifying vegetation free lava. If you’re also using it to determine the degree of weather which you are using to relate to the relative age of the lava flows, you should include this somewhere in the methods. It would also be useful if you went into a little more detail about why you are also looking for Hematite and Al(OH)3 to help identify these flows. If you are just using the 10/12 ratio, then what is the purpose of including these other ratios?
Response: By comparing the emissivity curve of sample WD-2w and the other four samples, it can explain that the B10/B12 ratio can be used for identifying whether the lavas covered by vegetation. What’s more, by comparing the emissivity curve of sample WD-3w and WD-3c (the depth of low emissivity near 9.6 µm for these samples is quite different). It can indicate that the B10/B12 ratio can be used to determine the weathering degree of lavas. We mentioned this point in the new manuscript. Considering that weathered volcanic lavas may contain Hematite and AL(OH)3, so we selected band ratios of B2/B1 and B4/B6 in the VNIR and SWIR regions displayed as green and blue respectively to highlight hematite and altered Al(OH)3 bearing minerals. Moreover, the band ratios of B2/B1 for displaying as green could highlight the area covered by snow and ice (the bright green color patterns on Figure 9), which is helpful to distinguish the lave flows from snow and ice.
Point 35: Line 249: Are you implying that this is the first case of contour mapping in this region? What about STRM, or ground based surveys?
Response 35: We did not mention that this research is the first case of contour mapping in this region. The resolution of ASTER DEM is 20 m, while the resolution of SRTM is 30m or 90m. In this paper, we try to demonstrate that the high quality of ASTER DEM data, which can almost provide global coverage on earth, is suitable for documenting the topographic and geomorphologic features of volcanoes.
Point 36: Line 253: Earlier you state an accuracy of 20m, is it 15 or 20? Where are you getting these numbers from?
Response 36: We are so sorry to make a mistake in the Table 2. The DEM data we used was the ASTER Level-3A product and derived from the ASTER Level-1A product, with vertical accuracy of 20 m. We have corrected it.
Point 37: Line 260: If these flows are already been mapped going back as early as 1979, how is this study novel? How does the spatial distribution different? What is the previous study you are referring to?
Response 37: The map in 1979 suggested the lava flows around Molabushan (M) belong to the Late Pleistocene, while our results demonstrated that they are at the same stage with the lava flows distributed around west flank of West Longmenshan (WL), which belong to the Holocene.
Another Response to Response 37: If this is your primary discovery, please expand on: “Our results basically consistent with the map in [27], except that they did not distinguish between the d and e phases [27].”
Response: After careful analysis and reading of relevant literature, we have modified the Results and Discussion section (refer the Response 33). Our results are basically consistent with the map in [27], except that they suggested the lava flows around Molabushan (M) belong to the Late Pleistocene, while our results demonstrated that they are at the same stage with the lava flows distributed around west flank of West Longmenshan (WL), which belong to the Holocene. We explained this point on page 14, line 303-307.
Another Response to Response 37: The color of this flow does match the WL flow, but M is also covered by vegetation. However, SWIR is very sensitive to vegetation. Do you have some evidence that this color is directly related to the flow and just that similar types of vegetation are growing of these two flows? If you are using some method to solve for this, it should be included in the methods.
Response: Figure 9 is an ASTER composite image which was taken on March 07, 2002, with band ratios of 10/12, 2/1 and 4/6 displayed as red, green and blue, respectively. There is no vegetation coverage in Wudalianchi region in March. The bright green color patterns on Figure 9 should be the area covered by snow and ice, not the vegetation. But the M is displaying as yellowish red color. Therefore, we suggest that this color is directly related to the lava flows, not vegetation.
Point 38: Line 269-276: There are valid questions. Consider attempting to answer these based on the data you have available.
Response 38: We have modified this paragraph clearly as shown in line 338-342 of page 16.
Point 39: Line 283: The DEMS can generate topographic features?
Response 39: We have changed “generate” into “provide”.
Best regards,
Yours sincerely,
Fu
Corresponding Author:
Prof. Bihong Fu,
Institute of Aerospace Information Research Institute, Chinese Academy of Sciences
No. 9 Dengzhuang South Road, Haidian District, Beijing 100094, P.R. China
Phone: +86-010-8217-8096
Fax: +86-10-8202-4251
E-mail: [email protected]

Reviewer 4 Report
Great effort by the authors in submitting this revised manuscript. However, minor revision is needed before the manuscript could be accepted for publication. The manuscript should be checked again for language editing and style. I have marked my comments in the manuscript.

Author Response
Dear Reviewer,
Thank you so much for your constructive suggestions and comments on our manuscript entitled “New insights of geomorphologic and lithologic features on Wudalianchi volcanoes in the Northeastern China from the ASTER multispectral data”, which submitted to the Special Issue "ASTER 20th Anniversary" of Remote Sensing. According to your suggestions and comments, we have modified the manuscript carefully as following.
Point 1: Page 2, Line 49: covered-covers.
Response 1: There, we used “covered” because that the capability of ASTER to measure in SWIR was until 2006. For now, the SWIR is no longer functioning.
Point 2: Page 8, Line 204: Needs to be modified!
Response 2: We have changed “The flow chart to describe the use of our datasets.” to “The flowchart of the datasets used in this study.” following your suggestion.
Point 3: Page 8, Line 207-211: A little more explanation on how contour image and 3D image were generated would be useful to the readers.
Response 3: A contour image was derived from ASTER Level-3A DEM data by using ER-Mapper (Earth Resource-Mapper) software. Input the DEM data, then set the suitable contour interval, and the contour map will be generated automatically. Three-dimensional (3D) perspective view image was generated through superimposing a three bands color composite image (the bands 2, 3, and 1 assigned in red, green and blue, respectively) on the ASTER DEM data by using the same software. Input the DEM data and the three bands color composite image at the same time, then set the suitable value of vertical exaggeration, the software will automatically generate the 3D image.
Point 4: Page 15, Line 369-373: Sentence is too long. Could be split into two sentences.
Response 4: We split this sentence into two sentences as “The most remarkable shield volcano with a diameter of 2 km is located in middle of image, which consists of three craters (EL, WL and X in Figure 7a). Two of them (WL and EL) appear one pair of nearly E-W extending glasses with a diameter of ca.400 m in middle east part, and crater of Xiaogushan (X on Figure 7a) in the south flank of EL crater appears as a northeastward-facing horseshoe shape with a diameter of 200 m”.
Point 5: Page 16, Line 397-398: needs to be modified as it didn't sound grammatically right.
Response 5: We have changed this sentence into “Laoheishao crater covered with green vegetation exhibits a cone and locates in the central of the Three-dimensional (3D) perspective view image (Figure 8)”.
Point 6: Page 16, Line 401-403: Any reference to support this statement?
Response 6: We have changed this sentence as following “Based on different color tones and texture characteristics displayed on remote sensing images, different lavas of Laoheishan and Huoshaoshan can be distinguished”.
Point 7: Page 16, Line 403-411: Content is fine but sentences need to be modified to sound alright.
Response 7: We modified these sentences as “Based on different color tones and texture characteristics displayed on remote sensing images, different lavas of Laoheishan and Huoshaoshan can be distinguished. Lava flows from Laoheishan crater can be classified into two types: one is the reddish grey lavas exhibited radial pattern and flowed as far as 15 km (indicated by red arrows on Figure 8), and another is the lavas in darkish grey to black just distributed around the crater (indicated by blue arrows on Figure 8). Lavas from Huoshaoshan crater in black color mainly flowed northward and southward (indicated by white arrows on Figure 8)”.
Point 8: Page 19, Line 487-488: Needs modification.
Response 8: According to your comment, we modified this sentence into “except for the different judgments of what stage that the lava flows around Molabushan (M) should belong to”.
Point 9: Page 19, Line 494-496: Sentences need modification.
Response 9: We modified these sentences as “Another point that needs to be concerned is the subtle differences of lava flows’ color tones among the Laoheishan and Huoshaoshan volcanoes. As concerning the different color pattern in ASTER image of these lava flows represent the TIR spectral differences, we suggest that at least two different aspects may have been responsible for subtle spectral differences:” accoeding to your suggestion.
Response 10: Other grammar and spelling mistakes have all been modified according to your comments. Please see the manuscript in detail.
Best regards,
Yours sincerely,
Fu
Corresponding Author:
Prof. Bihong Fu,
Institute of Aerospace Information Research Institute, Chinese Academy of Sciences
No. 9 Dengzhuang South Road, Haidian District, Beijing 100094, P.R. China
Phone: +86-010-8217-8096
Fax: +86-10-8202-4251
E-mail: [email protected]
Round 2
Reviewer 2 Report
Dear authors,
Thank you for delivering responses to my comments. I think your manuscript is ready and no further comments from my point of view. Please take notes that minor check for English and text is required.
This manuscript is a resubmission of an earlier submission. The following is a list of the peer review reports and author responses from that submission.
Round 1
Reviewer 1 Report
Firstly, I would like to congratulate the authors for having conducted a good research on Geomorphologic and Lithologic Features of a volcano using RS techniques. However, it needs some minor to moderate correction before it can be published. I have marked my comments on the manuscript. Few suggestions.
1) Some of the maps generated do not have spatial co-ordinates/Lat-Long
2) Methodology section is not clear or not enough. It is better to have a separate section on materials and methods.
3) Discussion could be improved a little more.

Reviewer 2 Report
Here you can find the review of the manuscript remotesensing-596070. The manuscript is conforming to the journal requirements and formats, and a suitable quality data have been included. Major recommendations are related to the data presentation and manuscript organisation. Authors are encouraged to take into account some others considerations too.
I hope the following comments can help authors to improve the present manuscript.
- A new manuscript organisation is proposed including a “Methodology” section and joining “Results” and “Discussion” ones. This new “Methodology” section is necessary in order to get a clearly view of what and how data have been used. Some paragraphs must be relocated at these two new proposed sections. I feel it could significantly improved this slightly blurry manuscript.
Introduction
- Figure 1a. A legend must be included with the volcanoes symbols. In general, in all figures, information as names, numbers, symbols, … must be explained in a legend.
- Figure 1b. The 14 volcanoes are not visible. Please, remark.
Geological setting
- Figure 2. The perspective and the subducting slab dip must be improved, and depth data must be included.
- A geological sketch showing the tectonic context/volcanism relation is necessary. Faults, lithologies, ages, …
- Define the different volcanic facies could be included, as well as the different alteration processes and the resulting rocks. One of the mail goal of the manuscript is the ASTER/lithologies relation, and the facies must be well documented.
- Figure 9. The photographs must be relocated at this section. Or others geologically representative photographs.
ASTER data and Results
- Figures 3b, 4. A colour bar is also necessary.
- What are shied-shaped or isolated cones must be identified.
- Figures 5, 7, 8. All information must be explained in the corresponding legend.
- Table 2. It must be cited and relocated before the Figure 6.
Discussion
- This section must be improved (and relocated) since it appears as a results presentation exclusively.
- Last, I recommend a native speaker English revision.
I urge the authors not to view this solely as a critique but a chance to improve this paper to convey their ideas and data in the best possible and most accurate manner.
Warm regards
Reviewer 3 Report
Review of the paper:
New Insights of Geomorphologic and Lithologic Features on Wudalianchi Volcanoes from the ASTER Multispectral Data
by Fu et al.,
General comments:
Remote sensing has been widely applied to volcanology and this study is an example that illustrates how remote sensing can be used for volcano application. In this work, the authors show the possibility to derive lithological map and distinguish lava flow unit based on ASTER multispectral and 3D perspective image, The result shows ASTER multispectral image can be used to determine relative dating of Quaternary volcanoes in the semi-arid region and for ASTER 3D perspective image can track the flow directions of different Holocene lava flow of Wudalianchi.
This study is quite interesting; however, there are some comments that should be addressed by the authors:
Major comments
The authors didn’t describe methods clearly. I recommend using flowchart to simplify the explanation also clear datasets that you were used. In your study the authors didn’t mention date of ASTER multispectral data that were used (only ASTER DEM), also what band are you using? VNIR? or all the 14 bands? Since the thermal spatial resolution is 90 m instead of 15 m. In section 3 you mention about ASTER data and results, I recommend you separate this section since ASTER is data that you were used, you didn’t describe your dataset and methods very well in the section. Then you must define your results in separate section. In Figure 5, How is the ASTER 3D perspective can show the flow direction? This not stated clearly in the method.
Minor comments
Ln 22 “molten lavas of Laoheishan and Huoshaoshan flows,”. Please rephrase, ex: Laoheishan and Huoshaoshan lava flow. Ln 130 “ASTER data and Reuslts”. Results? I suggest use lava flow instead of molten lava flow in all sentence.

Reviewer 4 Report
This research has potential, there are some useful figures within that really require further analysis. Beyond that, grammar corrections and language edits are necessary. There is also a need to reduce the amount of repetitive and superfluous information while including necessary information. For example, there is an entire section on geologic setting that is largely unrelated to the rest of the paper, but not methods section.
Line 14: It doesn’t “have” the capability to measure in SWIR, it “had” the capability until 2006
Line 26: imaging?
Line 34-35: remote sensing can provide a certain resolution of spectral, spatial, and temporal coverage based on the type of sensor, but saying it provides spectral, spatial and temporal coverage doesn’t have a clear meaning.
Line 42: TIMs is brought up here, is there a point? Consider removing.
Line 46-47: Covered, be clear SWIR is no longer functioning, correct throughout
Line 52: geologic applications is probably too broad here, the temperature alone from TM and ETM has geologic applications it just isn’t useful for determining composition. Be more specific.
Line 55-56: “Therefore…. surface.” this sentence is redundant with the previous sentence, combine or remove.
Line 57-58: “Especially” is not needed here; you are going straight for discussing emissivity in one sentence to temperature in the next without distinguishing between the two.
Line 58-61: This sentence needs to be re-written more clearly; now you are discussing the benefit of global coverage without establishing ASTER coverage, consider introducing this earlier in this paragraph
Line 62: name the volcanoes
Line 64: what do you mean “it” still lacks? The volcanoes? Research in general?
Figure 1: Both figures are lacking a color ramp or explanation of what they variations in grey represent.
Line 74-76: How does this field investigation relate to this study? This just seems tacked on.
Paragraph Lines 78-85: Parts of this paragraph are redundant with information in a previous paragraph (lines 62-66). They could probably be combined.
Line 96: this sentence is worded poorly and difficult to understand.
Section 2: This section is interesting, but it’s full of superfluous information that doesn’t help to support the methods, results, or conclusions. Consider condensing this into only one paragraph of critical information.
Figure 2: How is this a geodynamic model? What software was used? What inputs? This looks like it was just drawn and is not necessary based on any real information. How is this related to identifying the geomorphologic and lithological features of the volcanoes? Consider removing.
Line 130: Results
Table 1: This table is not needed, the acquisition timing and DEM specification all should have been included in a methods section.
Line 137: “It’s clear to see” is subjective, consider removing
Line 140-42: How does Figure 3 demonstrate that these volcanoes are shield or cone volcanoes? What indicating features are you using?
Line 151-170: How were these heights determined? Methods are needed.
Line 177-179: Sentence unclear, what is being done here?
Line 182: remarkable is subjective.
Line 183: What do you mean by “damaged”?
Line 184: “spectacular” is subjective, as is “clearly”
Line 188/189/Figure 5: what distinguishes the red arrow and blue arrow lava flows? Is there a border? Why do you see this difference?
Figure 5: What is the vertical exaggeration?
Line 211-213: What is the significance of this lava flow index? What are you trying to identify/index?
Figure 6/Table 2: Is the point of this table just to demonstrate what the average spectra of the lava flow are? If so, is there a need to include specific spectra rather than an averaged spectrum? Also, how were these spectra collected? In what lab? Averaged over how many scans? Are they continuum removed? Where they heated? Please include in methods.
Table 2: Which lava flow were they associated with? Can you add these points on a map? Can you be more descriptive in the lithology? Did you actually scan the region with the lichen in 2W? If so, what is the purpose of this?
Line 255-226: Is different eruptive stages the only possibility for these color patterns? Expand on what these different colors mean? Do they relate to changes in roughness? Composition? Vescularity? Alteration? What is the significance of figures 7 and 8 beyond just having different colors for different flows?
Lines 230-236: How do you make the connection of the different color flows tot eh different timing of flows? Is this compared against established maps of these flows? If so, what is the significance of these findings?
Figure 7 and 8: What is the significance of the numbers of this figure? It’s not clear how this relative dating was made, please expand.
Line 249: Are you implying that this is the first case of contour mapping in this region? What about STRM, or ground based surveys?
Line 253: Earlier you state an accuracy of 20m, is it 15 or 20? Where are you getting these numbers from?
Line 260: If these flows are already been mapped going back as early as 1979, how is this study novel? How does the spatial distribution different? What is the previous study you are referring to?
Line 269-276: There are valid questions. Consider attempting to answer these based on the data you have available.
Line 283: The DEMS can generate topographic features?
Round 2
Reviewer 3 Report
Dear Author,
Thank you for your response, however, there are some comments that are not clearly stated in your manuscript.
Point 2: In your study the authors didn’t mention date of ASTER multispectral data that were used (only ASTER DEM), also what band are you using? VNIR? or all the 14 bands? Since the thermal spatial resolution is 90 m instead of 15 m.
Response 2: The new Figure 8 and 9 are two ASTER composite images, which were taken on Match. 07, 2002 and June. 04, 2004, respectively. For both of them, we used band ratios of 10/12, 2/1 and 4/6 displayed as red, green and blue, since the band ratios of 10/12 could distinguish fresh lavas and weathered lavas, while the band ratios of 2/1 and 4/6 can be used to highlight hematite and altered Al-(OH)3 bearing minerals. We explained the derived date and band ratios of the ASTER data what we used in line 397 and 417, respectively.
Another Response to Response 2: It is still not clear why do you use different date (June. 04, 2004) instead of March. 07, 2002 to produce an enlarged image of lithological map? and also where is another data of June 9, 2002 in lithological results?
You still didn't describe well the use of your datasets, I reccomend to use flow chart or detail table that clearly describe the use of your datasets including the final output that you expect from each date of the data.
Point 4: In Figure 5, how is the ASTER 3D perspective can show the flow direction? This not stated clearly in the method.
Response 4: We distinguished the flow directions of lavas according to their texture characteristics and color tone. We have explained this in detail in line 378-384 of page 15.
Another Response to Response 4: This still not clearly describe, what texture and color tone did you consider to derive the direction of the flow? Is it only based from ASTER 3D? or do you also have field observation?
Reviewer 4 Report
Methods need to be expanded, still a lot of information missing from this section, such as how the lava flow maps were generated, Grammar corrections needed.
Point 4: Line 42: TIMs is brought up here, is there a point? Consider removing.
Response 4: We want to explain further the spectral differences of lava flows caused by the weathering can be measured and mapped with multispectral imaging system.
This is still out of place and isn’t really related to any other points made in the manuscript. If including this information is important to the manuscript, be clear within the manuscript why it is important.
Table 1 needs to be cited.
Point 12: Figure 1: Both figures are lacking a color ramp or explanation of what they variations in grey represent.
Response 12: We have added a color ramp in Figure 1 to explain what they variations in grey represent.
This should be a specific temperature, not just “high/low”
Point 13: Line 74-76: How does this field investigation relate to this study? This just seems tacked on.
Response 13: During the field investigation, we had observed the difference of vegetation coverage in the Wudalianchi volcanoes, which can provide useful information to understand the spectral difference of the lava flows in the Laoheishan and Huoshaoshan lava flows.
That’s a good reason, but this needs to be explained in the paper, as it is, as previously stated, it seems tacked on.
Point 17: Figure 2: How is this a geodynamic model? What software was used? What inputs? This looks like it was just drawn and is not necessary based on any real information. How is this related to identifying the geomorphologic and lithological features of the volcanoes? Consider removing.
Response 17: Current Figure 3 is a sketch map, which displays the relationship between the active intraplate volcanoes in NE China and the deep subduction of the Pacific slab. This 3D geodynamic model was drawn using Artificial Intelligence software based on analysis of the topographic image as well as geophysical evidence given by the previous studies. Additionally, we have improved the quality of figure according to the new data published by Wei et al (2018).
This should be explained in the paper, it’s still not entirely clear. You call it both a sketch map and a 3D model, was it generated using AI, or is it based on the results of the program?
“The DEM data was derived from the Level-3A ASTER data, with accuracy of 15 m”
This needs citation.
Point 25: Line 183: What do you mean by “damaged”?
Response 25: The “damaged” means the cone of Huoshaoshan is incomplete compared with the Laoheishan crater.
This is still unclear and has not been addresses in the manuscript.
Point 27: Line 188/189/Figure 5: what distinguishes the red arrow and blue arrow lava flows? Is there a border? Why do you see this difference?
Response 27: We analyzed flow direction of lavas according to the texture characteristics and color tone of the lava flow.
This does not answer the question and was not addressed in the manuscript.
Point 28: Figure 5: What is the vertical exaggeration?
Response 28: We exaggerated topographic features vertically in order to display clearly the morphologic features of these young volcanoes
This is an overlain DEM image, the DEM has some vertical exaggeration value, what is it.
Point 29: Line 211-213: What is the significance of this lava flow index? What are you trying to identify/index?
Response 29: We defined this lava flow index in order to identify the spectral subtle variations of these lava flows with different age in the Wudalianchi volcanoes.
This needs to be addressed in the manuscript, this description still does not clearly define what this lava flow index is or why it’s significant. It this what you’re calling the spectral library? Or the list of sample locations?
Point 30: Figure 6/Table 2: Is the point of this table just to demonstrate what the average spectra of the lava flow are? If so, is there a need to include specific spectra rather than an averaged spectrum? Also, how were these spectra collected? In what lab? Averaged over how many scans? Are they continuum removed? Where they heated? Please include in methods.
Response 30: Figure 6 (the new Figure 4) shows the laboratory emissivity in different bands of those samples. We use Figure 6 (the new Figure 4) to determine how to perform bands operations to highlight differences of spectral emissivity between different samples in order to better distinguish different types of samples.
In Nimoiya’s Spectral Lab. of Geological Survey of Japan, we measured the laboratory emissivity by using portable FTIR (Fourier Transform Infrared Spectrometer) spectrometer (µ-FITR, Model 102) manufactured by Designs and Prototypes, Ltd., USA. The emissivity measurements of the samples of relatively low temperature (typically < 100 degree in Celsius) at the natural surfaces are difficult in achieving high S/N ratio, so the measurements were generally made for the polished surfaces of the highly heated samples using the plate heating pot in the laboratory.
This needs to be included in the methods section of the paper and expanded to address the original questions.
Point 31: Table 2: Which lava flow were they associated with? Can you add these points on a map? Can you be more descriptive in the lithology? Did you actually scan the region with the lichen in 2W? If so, what is the purpose of this?
Response 31: We marked the sampling locations on Figure 8, which are associated with the lava flows of Laoheishan and Huoshaoshan volcanoes. We measured the weathered surface of sample WD-2w covered partially by the dried lichen as shown in Figure 4. The emissivity curve of sample WD-2w is displaying as flat one compared with that of fresh surface.
Point the reader to figure 8 in the description for table 2, the lichen explanation needs to be included in methods.
Point 32: Line 225-226: Is different eruptive stages the only possibility for these color patterns? Expand on what these different colors mean? Do they relate to changes in roughness? Composition? Vescularity? Alteration? What is the significance of figures 7 and 8 beyond just having different colors for different flows?
Response 32: In the Wudalianchi region, the lava rocks are the high-K (Potassium) basalt content with an average of 5.28% and average SiO2 (Silicondioxide) content of 50.46%. The mineral composition has not big difference. Therefore, we consider that both the facies and weathering differences of volcanic rocks in the different eruptive stages are responsible for these color patterns. During the field investigation, we had observed the difference in vegetation coverage, which can also affect these color patterns in the Wudalianchi area.
Not addressed in manuscript
Point 33: Lines 230-236: How do you make the connection of the different color flows to the different timing of flows? Is this compared against established maps of these flows? If so, what is the significance of these findings?
Response 33: We have compared this against the established maps in Reference [27]. Our interpretation for relative dating has a good agreement with geologic mapping given by [27], but the spatial distribution of these lava flows in different stages is different with that revealed by the previous study, for they did not distinguish between the d and e phases. Therefore, it is necessary to re-measure age for the lava flows in the Wudalianchi volcanic results revealed by this study. We explained this point in detail on page 16, line 402-414.
Not explained clearly
Point 34: Figure 7 and 8: What is the significance of the numbers of this figure? It’s not clear how this relative dating was made, please expand.
Response 34: The numbers 1-5 in Figure8-9 refer to the five lakes that make up the Wudalianchi Pool. And letters a-e in Figure 8 represent the relative age dating of lava rocks according to the color tone of the lava flows.
This does not explain how color tone relates to age.
Point 37: Line 260: If these flows are already been mapped going back as early as 1979, how is this study novel? How does the spatial distribution different? What is the previous study you are referring to?
Response 37: The map in 1979 had divided these lava flows into four phases, while our results show 5 stages. We explained this point in detail on page 16, line 402-414.
If this is your primary discovery, please expand on: “Our results basically consistent with the map in [3627], except that they did not distinguish between the d and e phases [3627].”